# Dominant role of DNA methylation over H3K9me3 for IAP silencing in endoderm

Zeyang Wang [1,9], Rui Fan[1,8,9], Angela Russo [1], Filippo M. Cernilogar [1], Alexander Nuber[1], Silvia Schirge[2,3], Irina Shcherbakova[1], Iva Dzhilyanova [1], Enes Ugur[4], Tobias Anton[4], Lisa Richter [5], Heinrich Leonhardt [4], Heiko Lickert [2,3,6,7] & Gunnar Schotta [1] ✉

Silencing of endogenous retroviruses (ERVs) is largely mediated by repressive chromatin modifications H3K9me3 and DNA methylation. On ERVs, these modifications are mainly deposited by the histone methyltransferase *Setdb1* and by the maintenance DNA methyltransferase *Dnmt1*. Knock-out of either *Setdb1* or *Dnmt1* leads to ERV de-repression in various cell types. However, it is currently not known if H3K9me3 and DNA methylation depend on each other for ERV silencing. Here we show that conditional knock-out of *Setdb1* in mouse embryonic endoderm results in ERV de-repression in visceral endoderm (VE) descendants and does not occur in definitive endoderm (DE). Deletion of *Setdb1* in VE progenitors results in loss of H3K9me3 and reduced DNA methylation of *Intracisternal A-particle* (*IAP*) elements, consistent with up-regulation of this ERV family. In DE, loss of *Setdb1* does not affect H3K9me3 nor DNA methylation, suggesting *Setdb1*-independent pathways for maintaining these modifications. Importantly, *Dnmt1* knock-out results in *IAP* de-repression in both visceral and definitive endoderm cells, while H3K9me3 is unaltered. Thus, our data suggest a dominant role of DNA methylation over H3K9me3 for *IAP* silencing in endoderm cells. Our findings suggest that Setdb1-mediated H3K9me3 is not sufficient for *IAP* silencing, but rather critical for maintaining high DNA methylation.

ERVs are remnants of retroviral germline integrations during evolution. In mammalian genomes, a large proportion of ERV Long Terminal Repeats (LTR) contribute to the physiological regulation of gene expression during development. In this sense, ERV initiated transcripts contribute to pluripotency regulation in both mice and human embryonic stem cells[1,2]. In contrast, aberrantly high activity of ERVs is associated with diseases and abnormal development. Dys-regulation of ERV LTRs can drive expression of oncogenes in human tumor cells[3–5]

and, overexpression of ERVs is a feature of autoimmune diseases[6]. Thus, silencing mechanisms, which restrict ERV activity are important to ensure proper development and, misregulation may lead to disease.

Silencing of endogenous retroviruses is mediated by heterochromatin and, in particular, by establishment of H3K9me3 and DNA methylation[7]. The major H3K9me3 specific histone methyltransferase (HMTase) for ERVs is SETDB1[8], but additional HMTases, such as SUV39H, SETDB2 and PRDM enzymes, can contribute to establishing

[1]Division of Molecular Biology, Biomedical Center, Faculty of Medicine, LMU Munich, Munich, Germany. [2]Helmholtz Zentrum München, Institute of Stem Cell Research, Neuherberg, Germany. [3]Helmholtz Zentrum München, Institute of Diabetes and Regeneration Research, Neuherberg, Germany. [4]Biozentrum, LMU Munich, Munich, Germany. [5]Biomedical Center (BMC), Core Facility Flow Cytometry, Faculty of Medicine, LMU Munich, Munich, Germany. [6]German Center for Diabetes Research (DZD), Neuherberg, Germany. [7]Technische Universität München, Munich, Germany. [8]Present address: Embryonic Self-Organization Research Group, Max Planck Institute for Molecular Biomedicine, Münster, Germany. [9]These authors contributed equally: Zeyang Wang, Rui Fan. ✉e-mail: gunnar.schotta@bmc.med.lmu.de

this modification[9–12]. Establishment of H3K9me3 on ERVs depends on the sequence-specific recognition by KRAB-ZFP proteins[13]. The KRAB domain of these proteins is bound by the corepressor TRIM28, which then recruits SETDB1[14]. DNA methylation on ERVs is deposited during preimplantation development and then maintained by DNMT1. Establishment and maintenance of DNA methylation on ERVs relates to the H3K9me3 pathway. This is shown by impaired DNA methylation in *Trim28* ko embryos[15] and upon deletion of *Setdb1* in ESCs and other cell types[16,17]. Mechanistically, the connection between H3K9me3 and maintenance of DNA methylation is not fully understood. UHRF1 can target DNMT1 to ERVs through binding of H3K9me3 and hemi-methylated DNA[18,19], however, UHRF1 mutant proteins with impaired H3K9me3 binding can still maintain substantial levels of DNA methylation[20].

Although H3K9me3 and DNA methylation are both enriched on a subset of ERVs, e.g. *IAP* elements, the role of these modifications for silencing seems to differ in various cell types. Deletion of *Setdb1* in embryonic stem cells (ESC) leads to strong *IAP* de-repression, whereas constitutive *Dnmt1* ko ESCs did not show clear transcriptional changes of *IAP* elements[21]. However, acute deletion of *Dnmt1* in ESCs did result in transient *IAP* de-repression, demonstrating that DNA methylation has an important function for *IAP* silencing in ESCs[22]. In differentiated cells, DNA methylation was initially found to play a crucial role for *IAP* silencing, as *Dnmt1* ko embryos display strong *IAP* expression in various cell types[23]. Specific impairment of DNA methylation in neuronal cells could recapitulate these findings[24,25]. Interestingly, deletion of *Setdb1* in neuronal cells also results in *IAP* de-repression[26], although a broader investigation of the roles of *Setdb1* in differentiated cells revealed that *Setdb1* is dispensable for *IAP* silencing in some differentiated cell types[27]. These findings demonstrate that both modification pathways have important roles for *IAP* repression in distinct differentiated cell types. However, the interplay between these pathways has not been investigated in the same cell type.

Here, we investigate the role of *Setdb1* for ERV silencing upon germ layer differentiation in the endoderm lineage. We find that *Setdb1*-mediated ERV repression is restricted to extra-embryonic VE cells and does not occur in embryonic DE, suggesting ontogenesis-dependent regulatory mechanisms for ERV silencing. In both endoderm cell types, DNA methylation plays a dominant role for ERV and, in particular, *IAP* repression. Interestingly, H3K9me3 is maintained on de-repressed *IAP* elements in *Dnmt1* ko VE and DE cells, suggesting that H3K9me3 in absence of DNA methylation is not sufficient to establish transcriptional repression.

## Results

### Loss of *Setdb1* in embryonic endoderm cells leads to developmental defects

*Setdb1* is highly expressed during mouse embryonic development (Supplementary Fig. 1a). To study the role of *Setdb1* specifically in endoderm lineage development we combined a conditional *Setdb1*^flox allele with the *Sox17-2A-iCre* knock-in allele which expresses Cre recombinase in *Sox17* expressing endoderm cells[28]. *Sox17* is expressed in both embryonic and extraembryonic endoderm cells, therefore deletion of *Setdb1* is expected to occur in both lineages (Supplementary Fig. 1b). As the conditional deletion of one allele of *Setdb1* in the endoderm was phenotypically normal, we assign *Setdb1*^flox/+; *Sox17-2A-iCre* or *Setdb1*^flox/+ mice as control and mutant *Setdb1*^flox/flox; *Sox17-2A-iCre* as *Setdb1*^END.

*Setdb1*^END embryos do not display notable differences in development until embryonic day 8.5, where mutant embryos cannot complete turning (Fig. 1a). At later developmental stages (E9.0) the posterior part of *Setdb1*^END embryos deteriorates (Fig. 1a), followed by death of mutant embryos during later stages of pregnancy. The expression of key endoderm transcription factors *Foxa2* and *Sox17* was unaltered in E7.5 embryos (Fig. 1b), and no obvious structural

aberrations could be detected in E8.0 embryos (Supplementary Fig. 1c), demonstrating proper initiation of endoderm differentiation. Later developmental stages clearly show endoderm-derived gut tube structures (Fig. 1c, red arrows). In *Setdb1*^END embryos gut tube structures were not fully connected to neighboring tissue (Fig. 1c, red arrows), which could explain the turning defect. We did not observe notable differences in proliferation or apoptosis in *Setdb1*^END embryos (Supplementary Fig. 1d, e). Together our data demonstrate a crucial role for *Setdb1* in embryonic endoderm differentiation. In contrast to ESCs, where deletion of *Setdb1* results in cell death[8,29,30], *Setdb1*-deficient endoderm cells (between E7.5 - E8.5) do not display obvious viability or proliferation problems, but rather have functional defects which result in altered tissue integrity.

### *Setdb1*^END embryos display ERV de-repression specifically in visceral endoderm cells

To study transcriptional changes upon *Setdb1* deletion in endoderm cells, we combined *Setdb1*^flox; *Sox17-2A-iCre* alleles with an *EGFP-Cre* reporter allele[31]. The resulting *Setdb1*^flox; *Sox17-2A-iCre*; *EGFP-reporter* embryos displayed EGFP signals in *Sox17* expressing cells, whereas no *EGFP* expression was detected in *Setdb1*^flox; *EGFP-reporter* embryos which lack *Cre* activity at E8.5 (Supplementary Fig. 2a, b). The specific *Cre* reporter activity allowed us to FACS-isolate *EGFP* positive endoderm cells from control and *Setdb1*^END embryos (Supplementary Fig. 2c). Expression of *Setdb1* was strongly reduced in *Setdb1*^END endoderm cells (Supplementary Fig. 2d), demonstrating efficient deletion of *Setdb1*. Transcriptional profiling of control vs. *Setdb1*^END endoderm cells revealed significant up-regulation of 166 genes and down-regulation of only four genes (Fig. 2a, Supplementary Data 1). Top upregulated are germline-specific genes, known targets of *Setdb1* also in other cell types and consistent with the role of *Setdb1* in gene repression[17,21]. The top downregulated gene, *Nepn*, is an important marker gene for endoderm development[32,33]. Reduced *Nepn* expression domain could be confirmed by in situ hybridization (Supplementary Fig. 2e) and further indicates defective endoderm development observed in *Setdb1*^END embryos.

To investigate the role of *Setdb1* in ERV silencing we analyzed the expression of ERV families in control vs. *Setdb1*^END ex vivo endoderm cells. We observed strong de-repression of several ERV families (Fig. 2b, Supplementary Data 2). Expression of *LINE* elements was largely unchanged with only *L1Md_T* upregulated (Supplementary Fig. 2f). A prominent ERV family which is targeted by *Setdb1* in different cell types is *IAPEz*. Interestingly, *IAPEz* transcripts retained coding potential and, therefore, expression of *IAPEz* can be detected by presence of capsid protein GAG[34]. To investigate if ERV de-repression occurs in all embryonic endoderm lineage cells in *Setdb*^END embryos, we performed immunofluorescence analysis for IAPEz GAG. In E8.0 embryos we observe strong labeling of embryonic endoderm cells with the *EGFP* reporter, however, only a subset of reporter positive cells displayed GAG labeling (Fig. 2c). This suggests that *Setdb1* knock-out results in IAPEz de-repression in only a subset of embryonic endoderm cells. Embryonic endoderm is ontogenetically derived from both definitive endoderm which emerges around E6.5/E7.0 during gastrulation and visceral endoderm, which forms before gastrulation (Supplementary Fig. 1b). Since visceral endoderm cells assume morphology and function of definitive endoderm cells during gastrulation, they can only be distinguished using specific markers at the beginning of gastrulation. Therefore, to test if *IAPEz* de-repression may be specific to definitive or visceral endoderm in vivo, we stained E7.5 embryos for IAPEz GAG together with AFP, a specific marker of visceral endoderm[35]. AFP is strongly expressed in the visceral endoderm of the proximal extra-embryonic part of the embryo at E7.5 (Fig. 2d, region above white dashed line). Visceral endoderm descendants in the embryonic part start losing AFP expression during gastrulation, but we could still detect several AFP positive cells in the embryonic endoderm region at

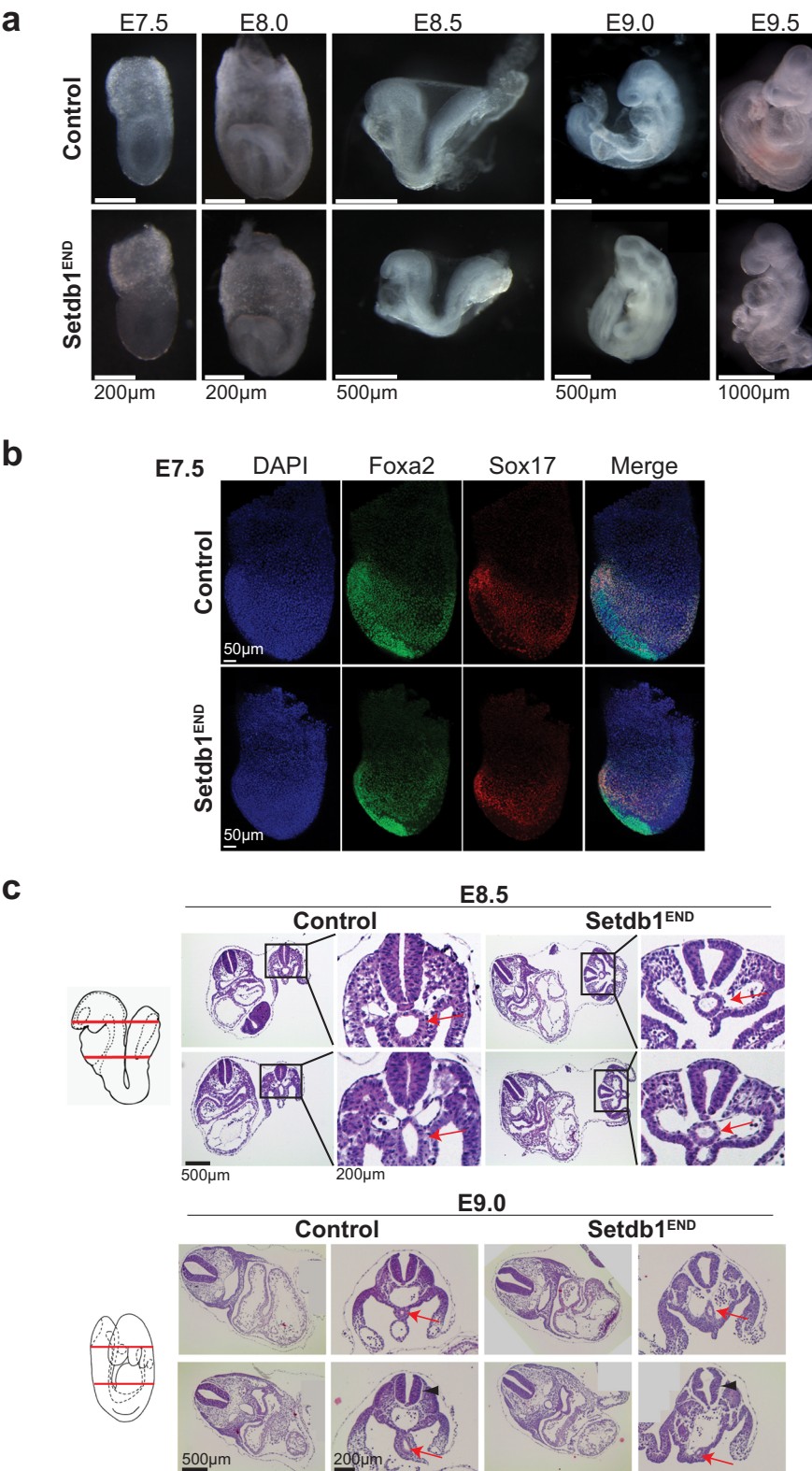

**Fig. 1 | Loss of Setdb1 in endoderm leads to strong developmental defects during embryogenesis. a** Lateral view of control and Setdb1[END] embryos at E7.5 (late bud stage), E7.75 (head fold stage), E8.5 (-6 somite) and E9. No visible developmental defects can be detected in Setdb1[END] embryos from E7.5 to E8.0. Setdb1[END] embryos show an axis turning defect which manifests from E8.5 and leads to strong posterior truncation at E9.0. Representative images from $n = 3$ per genotype and stage. **b** Lateral view of E7.5 control and Setdb1[END] embryos stained with Foxa2 and Sox17 antibodies (anterior to the left). The presence of both markers indicates that endoderm cells could be formed in Setdb1[END] embryos. Representative images from $n = 3$ per genotype and stage. **c** Hematoxylin/Eosin staining of transverse sections of E8.5 and E9.0 control and Setdb1[END] embryos. The approximate positions of the sections are indicated in the schematic. The black rectangle marks the region used for magnification. Red arrows indicate the hindgut region. The black arrowheads mark the neural tube. Representative images from $n = 3$ per genotype and stage.

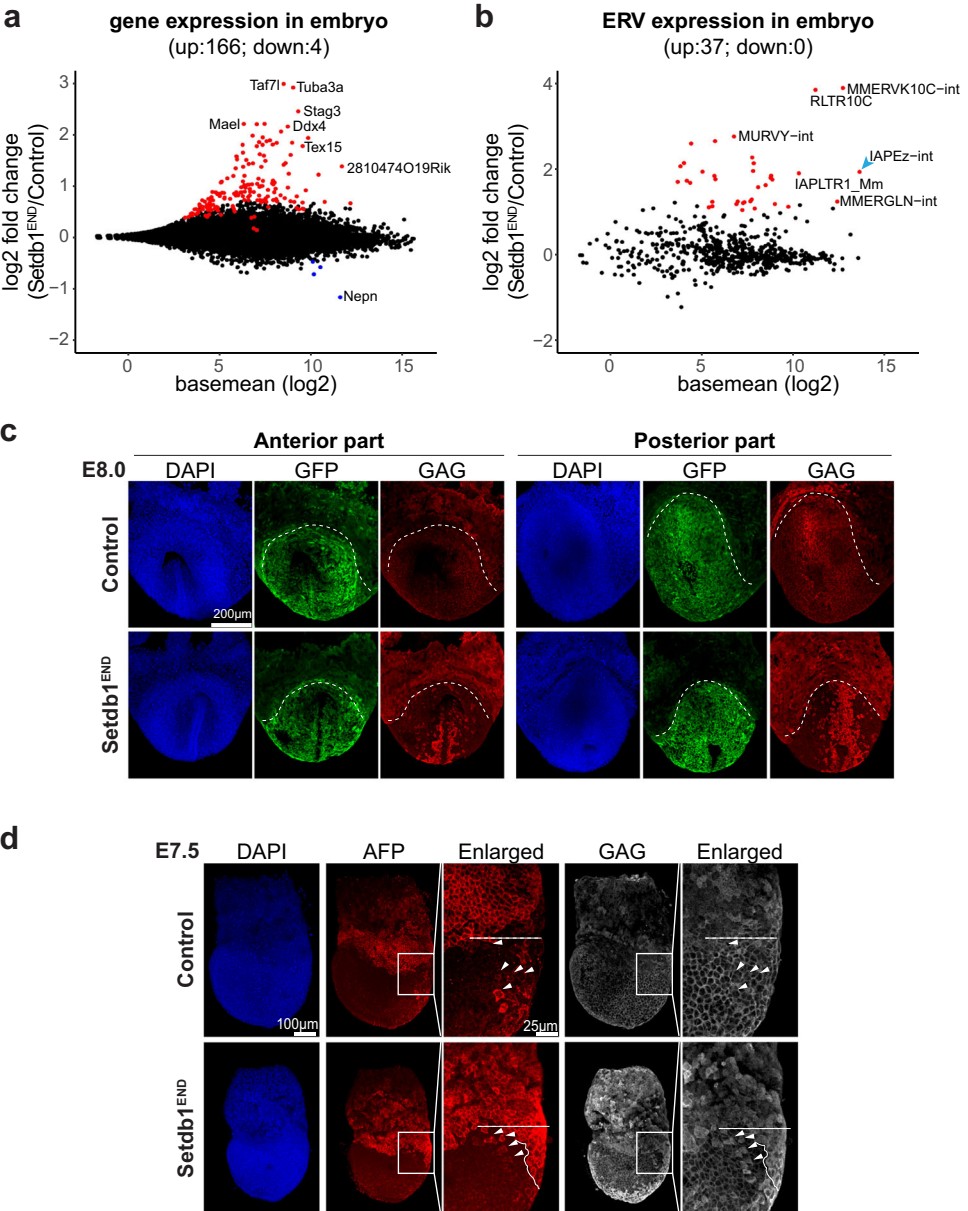

**Fig. 2 | Loss of Setdb1 leads to selective de-repression of IAP elements in visceral endoderm cells. a** Dot plot showing basemean expression vs. log2-fold change of protein coding genes in embryonic endoderm cells. Genes with significantly changed expression (adjusted $p$ value <0.01; $n = 3$ for each condition) are colored (red = increased expression in Setdb1$^{END}$ cells, blue = reduced expression in Setdb1$^{END}$ cells). Selected genes are labeled. **b** Dot plot showing basemean expression vs. log2-fold change of ERV families in embryonic endoderm cells. ERV families with significantly changed expression (Wald test with Benjamini–Hochberg correction, adjusted $p$ value <0.01, fold change >2; $n = 3$ for each condition) are colored (red = increased expression in Setdb1$^{END}$ cells, blue = reduced expression in Setdb1$^{END}$ cells). Selected ERV families are labeled. **c** Whole mount immunostaining of control and Setdb1$^{END}$ embryos using GFP (to detect Cre reporter activity) and IAP-GAG antibodies. Strong expression of IAP-GAG can only be detected in a subpopulation of endoderm cells. Dashed lines indicate the border between extra-embryonic and embryonic part. Representative images from $n = 3$ per genotype and stage. **d** Lateral view of E7.5 embryos stained with AFP (to mark visceral endoderm cells) and IAP-GAG antibodies (anterior to the left). The boxed regions indicate the positions of the enlargements. Dashed lines indicate the border between extraembryonic and embryonic part. White arrowheads indicate AFP expressing cells which are integrated in the embryonic endoderm region. In Setdb1$^{END}$ embryos, these cells display clear IAP-GAG staining. Representative images from $n = 3$ per genotype and stage.

E7.5 (Fig. 2d, region below white dashed line, cells marked by arrowheads). Notably, we detected clear IAPEz GAG expression in these AFP positive cells, whereas most AFP negative DE cells did not display GAG expression. These data suggest that *Setdb1* knockout results in specific *IAPEz* de-repression in the visceral, but not definitive endoderm.

**ERV de-repression in *Setdb1*-deficient visceral endoderm progenitors in vitro**
To investigate the molecular mechanisms underlying *Setdb1*-dependent ERV de-repression in visceral vs. definitive endoderm cells, we employed an in vitro differentiation system (Fig. 3a). ESCs from control and *Setdb1$^{END}$* mice were stimulated with Wnt3a and Activin to induce definitive endoderm cells[36]. *Gata6* overexpression in these ESCs triggered differentiation to extraembryonic endoderm (XEN), a progenitor stage of visceral endoderm cells[37]. Both, DE and XEN cells show expression of the *EGFP Cre* reporter (Supplementary Fig. 3a) and efficiently delete *Setdb1* (Supplementary Fig. 3b, c). In support of our hypothesis, that *Setdb1*-dependent de-repression of ERVs mainly occurs in the extra-embryonic endoderm lineage, we observed strong expression of IAP GAG in *Setdb1$^{END}$* XEN cells, but not in *Setdb1$^{END}$* DE

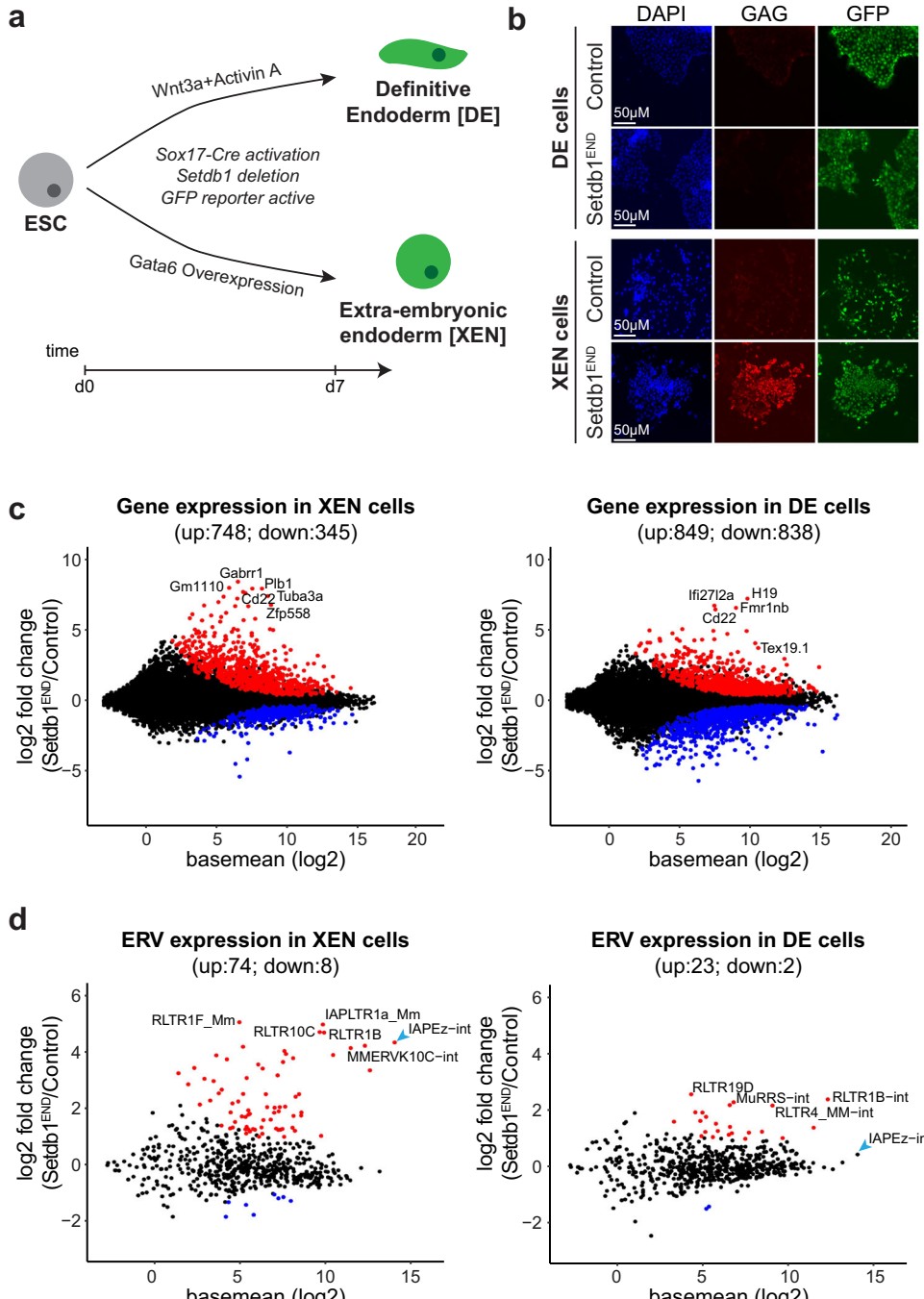

**Fig. 3 | Setdb1 is critical for IAP silencing in visceral endoderm progenitors, but not in definitive endoderm cells. a** Schematic of in vitro endoderm differentiation of control and Setdb1[END] ESCs. Definitive endoderm cells were generated by Wnt3a/ Activin stimulation. Visceral endoderm progenitor (XEN) cells were generated by overexpression of Gata6. **b** Immunofluorescence staining of in vitro differentiated control and Setdb1[END] endoderm cells. Cells were stained after 7 days of in vitro differentiation using IAP-GAG (marks IAP de-repression) and GFP (marks Cre reporter activity) antibodies. Strong IAP de-repression is only observed in Setdb1[END] XEN cells. Representative images from *n* = 3 experiments. **c** Dot plot showing basemean expression vs. log2-fold change of protein coding genes in in vitro

differentiated control vs. Setdb1[END] XEN and DE cells. Genes with significantly changed expression (Wald test with Benjamini–Hochberg correction, adjusted *p* value < 0.01; *n* = 3 for each condition) are colored (red = increased expression in Setdb1[END] cells, blue = reduced expression in Setdb1[END] cells). Selected genes are labeled. **d** Dot plot showing basemean expression vs. log2-fold change of ERV families in in vitro differentiated control vs. Setdb1[END] XEN and DE cells. ERV families with significantly changed expression (Wald test with Benjamini–Hochberg correction, adjusted *p* value < 0.01; *n* = 3 for each condition) are colored (red = increased expression in Setdb1[END] cells, blue = reduced expression in Setdb1[END] cells). Selected ERV families are labeled.

cells (Fig. 3b). To characterize *Setdb1* dependent transcriptional changes, we isolated differentiated control and *Setdb1* deficient DE and XEN cells for RNA-seq. PCA analysis revealed clear clustering of ESCs, DE and XEN cells (Supplementary Fig. 3d). Examination of control genes for ESCs and endoderm markers confirmed efficient

differentiation of both control and *Setdb1[END]* cells (Supplementary Fig. 3e). We detected many up- and down-regulated genes in both *Setdb1[END]* XEN and DE cells (Fig. 3c, Supplementary Data 3, 4).

When we analyzed ERV expression changes, we could detect strong de-repression of ERV classes in XEN cells, but only minor

expression changes in DE cells (Fig. 3d, Supplementary Data 5, 6). *LINE* elements did not display strong expression changes (Supplementary Fig. 3f). *IAPEz* elements which we found de-repressed in visceral endoderm cells in *Setdb1*[END] embryos did show strong de-repression in *Setdb1*[END] XEN cells, but no change was observed in *Setdb1*[END] DE cells (Fig. 3d, Supplementary Fig. 3g), even at a longer time period post *Setdb1* deletion (Supplementary Fig. 3h). Thus, in vitro differentiated XEN and DE cells reproduce the differential requirements for *Setdb1*-dependent ERV silencing, as observed in *Setdb1*[END] embryos.

## Impaired H3K9me3 and DNA methylation on IAPEz elements specifically in *Setdb1*[END] XEN cells

Next, we aimed to investigate whether selective changes in repressive chromatin modifications might explain the differential response of XEN or DE cells to *Setdb1* loss. To investigate this question, we generated H3K9me3 ChIP-seq data from control and *Setdb1*[END] XEN and DE cells. The appearance of H3K9me3 distribution was strikingly different in DE vs XEN cells. In particular, we detected a number of large megabase-size H3K9me3 domains in DE cells, which were not present in XEN cells (Supplementary Fig. 4). In *Setdb1*[END] DE cells, most of these regions were not compromised, suggesting that other H3K9me3-specific HMTases maintain these regions. Interestingly, in XEN cells, deletion of *Setdb1* resulted in appearance of large H3K9me3 domains, which were not present in control DE or XEN cells (Supplementary Fig. 4). Together these data suggest that alterations in the balance of H3K9me3 specific HMTases can lead to large-scale changes in the genome-wide distribution of this modification. To investigate to which extent changes in H3K9me3 would relate to gene expression changes, we identified peaks which lose H3K9me3 in *Setdb1*[END] cells. In *Setdb1*[END] DE cells, we detected 722 peaks with lost H3K9me3 signal, 84 of which occurred in the vicinity of regulated genes (Supplementary Fig. 5a). Only a small set of genes was upregulated and would suggest a repressive role for H3K9me3. For example, *Triml2* is marked by *Setdb1*-dependent H3K9me3 in DE cells and loss of H3K9me3 coincided with de-repression of *Triml2*. In XEN cells, *Triml2* was not modified by H3K9me3 and expression did not change between control and *Setdb1*[END] cells (Supplementary Fig. 5b). In XEN cells, we detected 17438 peaks with *Setdb1*-dependent H3K9me3, of which 614 were in the vicinity of regulated genes (Supplementary Fig. 5a). The majority of H3K9me3 marked genes was upregulated in *Setdb1*[END] cells, suggesting a role for *Setdb1* in gene repression. For example, *Gabrr1* is H3K9me3 modified in both DE and XEN cells, but only in XEN cells, we detected loss of H3K9me3 and de-repression of *Gabrr1* (Supplementary Fig. 5c).

Next, we investigated H3K9me3 changes specifically on ERV families. In *Setdb1*[END] XEN cells, we observed reduced H3K9me3 levels in several ERV families, including many upregulated ERVs (Fig. 4a, left panel). In contrast, H3K9me3 was reduced on very few ERV families in *Setdb1*[END] DE cells (Fig. 4a, right panel). H3K9me3 was unaltered on *LINE* elements in both *Setdb1*[END] XEN and DE cells (Supplementary Fig. 6a). Cumulative coverage analysis on *IAPEz* elements revealed that H3K9me3 was completely lost in *Setdb1*[END] XEN cells, whereas no difference could be observed in *Setdb1*[END] DE cells (Fig. 4b). These data suggest that *Setdb1* is the major H3K9me3 HMTase for *IAPEz* elements in XEN cells and, that other HMTases compensate for the loss of *Setdb1* in DE cells to maintain H3K9me3 on *IAPEz* elements and other ERVs.

Reduced H3K9me3 often correlates with reduced maintenance of DNA methylation[8,17]. To determine if reduced H3K9me3 in *Setdb1*[END] XEN cells would compromise DNA methylation, we measured DNA methylation levels specifically on *IAPEz* and *LINE1* elements using locus-specific bisulfite sequencing. Control ESCs as well as XEN and DE cells display high DNA methylation levels across *IAP-LTR* and *IAP-GAG* regions (Fig. 4c). *LINE1* elements only showed moderate DNA methylation in ESCs and XEN cells, but full methylation in DE cells (Supplementary Fig. 6b). Upon loss of *Setdb1*, DNA methylation is only affected in XEN cells, where we detected reduced levels across the *IAP-GAG*

region (Fig. 4c). These data agree with current models that maintenance of DNA methylation on repressive chromatin regions is coupled with the presence of H3K9me3[15–17]. Further, our data suggest that reduced H3K9me3 and DNA methylation allow higher transcriptional activity of ERVs in *Setdb1*[END] XEN cells. It is interesting to note that DNA methylation reduction on *LINE1* elements does not coincide with strongly reduced H3K9me3 (Supplementary Fig. 6). Perhaps, *Setdb1* is required for recruitment of de novo methylation by *Dnmt3a/b* during differentiation and other H3K9me3 HMTases could deposit H3K9me3 in absence of *Setdb1* on these elements.

In DE cells, SETDB1 localizes to *IAP* elements (Supplementary Fig. 7a), indicating that SETDB1 could mediate H3K9me3 in this cell type. However, loss of *Setdb1* in DE cells even for extended time periods did not result in noticeable *IAPEz* de-repression (Supplementary Fig. 7b). In vitro differentiation allows DE cells to proliferate until around day 14 (Supplementary Fig. 7c), which would allow for passive loss of H3K9me3 upon *Setdb1* deletion, but we could not detect reduced H3K9me3 on *IAPEz* elements at day 12 or day 14 (Supplementary Fig. 7d). These data suggest that H3K9me3 is maintained by other histone methyltransferases. We assessed whether *Suv39h* enzymes would be responsible for H3K9me3 deposition in DE cells. In *Suv39h* dko DE cells, we detected largely unaltered H3K9me3, DNA methylation and ERV transcription (Supplementary Fig. 8), suggesting a minor role for IAP regulation. However, it is still possible that *Suv39h* enzymes could compensate for the loss of *Setdb1*, or that other H3K9me3 HMTases could mediate H3K9me3 in DE.

## Loss of *Dnmt1* leads to ERV de-repression in both DE and XEN cells in presence of H3K9me3

To investigate if maintenance of repressive chromatin in *Setdb1*[END] DE cells prevents ERV de-repression we used *Dnmt1* knock-out ESCs to study the effect of impaired DNA methylation on ERV activity. *Dnmt1* ko ESCs and genetic background matched wildtype ESCs were in vitro differentiated to XEN and DE cells, respectively. We then performed RNA-seq analysis to determine transcriptional changes in *Dnmt1* ko XEN and DE cells. PCA analysis showed clear clustering of ESCs, DE and XEN cells (Supplementary Fig. 9a). Expression of specific marker genes revealed that *Dnmt1* ko ESCs efficiently differentiate to XEN and DE cells (Supplementary Fig. 9b). Transcriptional changes of coding genes between *Dnmt1* ko XEN and DE cells were observed (Supplementary Fig. 9c, Supplementary Data 7–9), with little overlap to transcriptional changes observed in *Setdb1*[END] cells. We then investigated transcriptional changes of ERV and LINE families in response to *Dnmt1* ko (Supplementary Fig. 9d, e, Supplementary Data 10–12). Undifferentiated *Dnmt1* ko ESCs did not show elevated *IAPEz* expression (Supplementary Fig. 9d), in agreement with previous studies[21]. Importantly, loss of *Dnmt1* resulted in strongly reduced DNA methylation (Supplementary Fig. 10) and *IAPEz* de-repression in both XEN and DE cells (Supplementary Fig. 9d). These data demonstrate that DNA methylation is critical for ERV silencing in both endoderm lineages. To test if upregulated *IAPEz* expression in *Dnmt1* ko endoderm cells would be due to impaired H3K9me3, we performed ChIP-seq analyses for this modification in control and *Dnmt1* ko ESCs, XEN and DE cells. In ESCs, H3K9me3 was not affected on IAP regions (Fig. 5a, b), suggesting that maintained H3K9me3 could support *IAP* silencing in ESCs. However, in both *Dnmt1* ko XEN and DE cells H3K9me3 was also maintained on *IAP* sequences, although *IAPEz* elements were strongly de-repressed (Fig. 5a, b). These data were supported by ChIP-qPCR analyses for H3K9me3 in wild type and *Dnmt1* ko XEN and DE cells, where we failed to detect striking changes in H3K9me3 on *IAPEz* regions, although other control regions, such as *H19* and *Polrmt* could display reduced signals (Supplementary Fig. 9f). Since *IAPEz* elements show little polymorphisms, small read mapping to unique elements is challenging and we cannot be sure which *IAPEz* insertions become transcriptionally active while maintaining H3K9me3 in *Dnmt1* ko DE

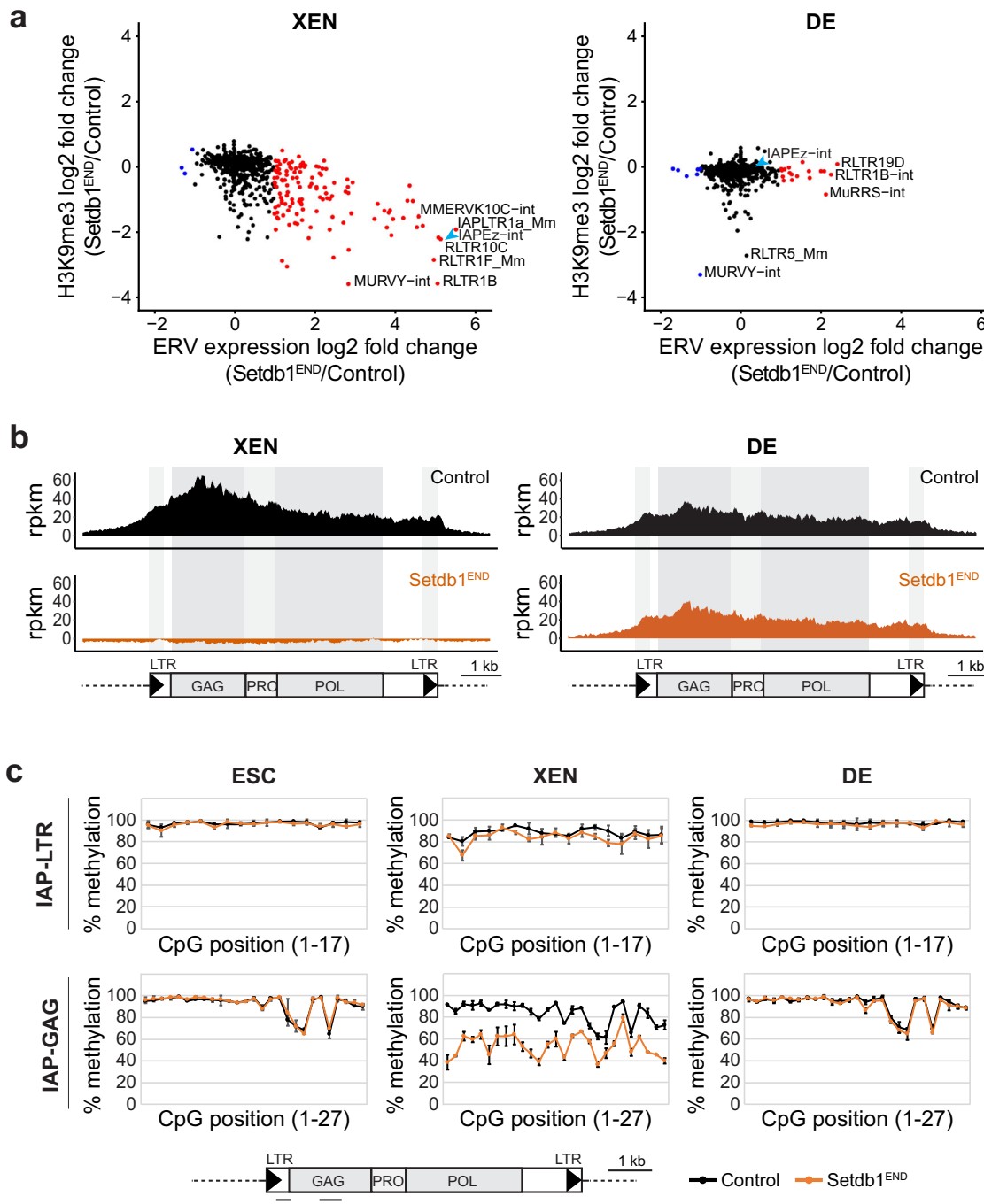

**Fig. 4 | Setdb1 is critical to maintain H3K9me3 and DNA methylation on ERVs in XEN, but not DE cells. a** Dot plot showing expression vs. H3K9me3 changes on ERV families between control and Setdb1$^{END}$ XEN and DE cells. ERV families with significantly changed expression (fold change > 2; $n$ = 3 for each condition) are colored (red = increased expression in Setdb1$^{END}$ cells). Selected ERV families are labeled. **b** Cumulative H3K9me3 ChIP-seq coverage across IAP elements in control and Setdb1$^{END}$ XEN and DE cells. The structure of IAP elements is shown schematically. (rpkm = reads per kilobase per million of reads). **c** Bisulfite-PCR analysis for DNA methylation in IAP-LTR and IAP-GAG regions. Positions of the PCR products are indicated in the schematic. Plots display the percentages of DNA methylation in individual CpG positions of IAP-LTR and IAP-GAG PCR fragments. DNA methylation analysis was performed in control and Setdb1$^{END}$ ESCs, XEN and DE cells. Error bars depict standard deviation ($n$ = 2; 500 sequences each). Source data are provided as a Source Data file.

and XEN cells. However, some *IAPEz* elements lack proper transcriptional termination at their 3'LTRs and, H3K9me3 presence can be detected in uniquely mapping regions neighboring the *IAPEz* insertion. Using this approach, we could identify examples of individual *IAPEz* elements that displayed significant transcriptional activity only in *Dnmt1* ko DE or XEN cells while maintaining H3K9me3 (Fig. 5c). Based on these data we conclude that DNA methylation is critical for IAP repression in endoderm lineages and, that the presence of H3K9me3 is

not sufficient to establish a repressive chromatin environment across these elements.

## Discussion

In this study, we have delineated the roles of H3K9me3 and DNA methylation for ERV regulation upon early embryonic vs. extra-embryonic endoderm development and differentiation. Our data demonstrate ontogenesis-dependent regulatory mechanism for IAP

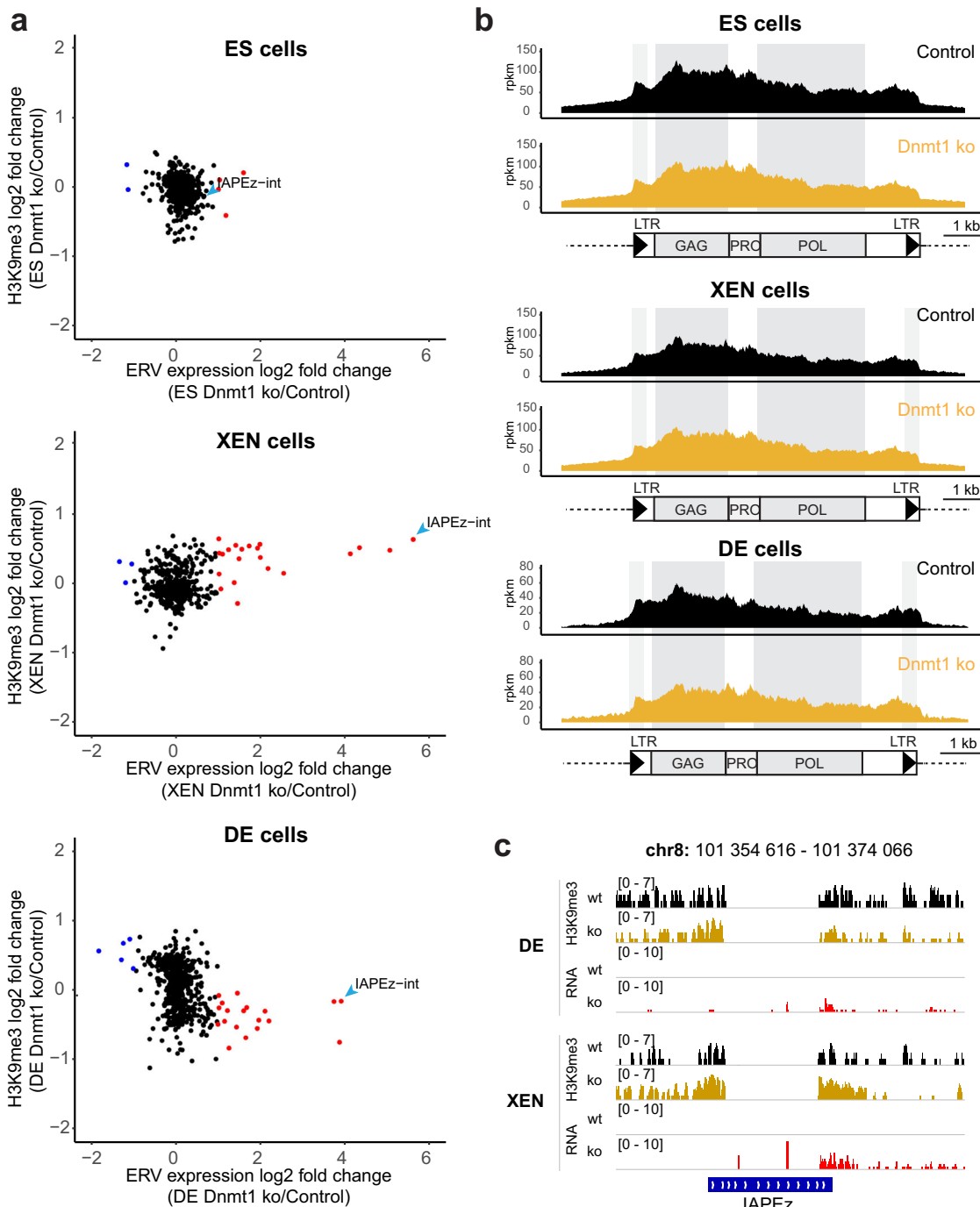

**Fig. 5 | Loss of Dnmt1 leads to IAP de-repression in both XEN and DE cells, while H3K9me3 is maintained. a** Dot plot showing basemean expression vs. H3K9me3 changes of ERV families in ESCs and in vitro differentiated wild type (J1) vs. Dnmt1 ko XEN and DE cells. ERV families with significantly changed expression (Wald test with Benjamini–Hochberg correction, adjusted *p* value < 0.01, fold change >2; *n* = 2 for each condition) are colored (red = increased expression in Dnmt1 ko cells, blue = reduced expression in Dnmt1 ko cells). Selected ERV families are labeled. **b** Cumulative H3K9me3 ChIP-seq coverage across IAP elements in control and Dnmt1 ko ES, XEN and DE cells. The structure of IAP elements is shown schematically. (rpkm = reads per kilobase per million of reads). **c** Improper transcriptional termination allows identification of individual IAPEz integrations with detectable expression in an H3K9me3 context. Genomic screenshot of an IAPEz integration with H3K9me3 coverage in uniquely mapping regions bordering the IAPEz sequence. In Dnmt1 ko DE and XEN cells, expression from this IAPEz element can be detected by presence of RNA in the uniquely mapping 3′ region of this element, which is likely due to improper transcriptional termination in the 3′LTR. H3K9me3 is unaltered in this region in Dnmt1 ko cells.

silencing in early endoderm development. In visceral endoderm cells, *Setdb1* is crucial to maintain H3K9me3 and DNA methylation on *IAP* elements. In definitive endoderm cells, H3K9me3 is not lost upon *Setdb1* deletion, suggesting compensation by other histone methyltransferases. This would be in agreement with a previous report that demonstrated reduced H3K9me3 only upon triple deletion of *Setdb1*,

*Suv39h1* and *Suv39h2* in definitive endoderm-derived liver cells[38]. We could not detect reduced H3K9me3 and IAP-GAG expression in *Suv39h* dko definitive endoderm cells. This would suggest that *Suv39h* enzymes do not play a major role for mediating H3K9me3 on *IAP* in early definitive endoderm. However, because of the maintained H3K9me3 on *IAP* in *Setdb1*[END] DE cells, *Suv39h* enzymes or other histone

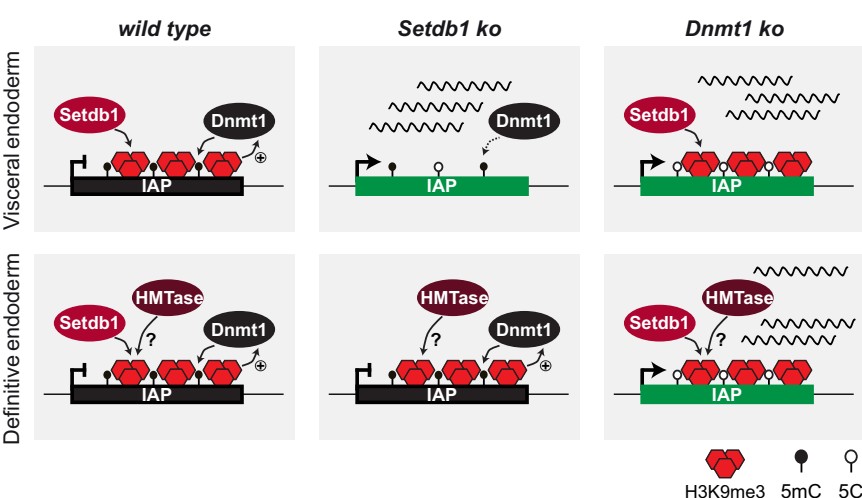

**Fig. 6 | Model.** Schematic depicts a dominant role of DNA methylation over H3K9me3 for ERV silencing in endoderm differentiation. Depletion of Setdb1 leads to augmented H3K9me3 and DNA methylation and impaired ERV silencing in visceral endoderm but not in definitive endoderm cells. In contrast, depletion of Dnmt1 impairs ERV repression in both visceral endoderm and definitive endoderm cells, although H3K9me3 is maintained. Together, our data suggest redundant cell type specific H3K9me3 maintenance pathways and a dominant role of DNA methylation for IAP silencing in endoderm cells.

methyltransferases (*Setdb2*, *Prdm* enzymes) may compensate the loss of *Setdb1*.

Our data suggest a dominant role of DNA methylation over H3K9me3 for *IAP* silencing (Fig. 6). Loss of DNA methylation resulted in *IAP* de-repression in both visceral and definitive endoderm. Surprisingly, H3K9me3 on *IAP* elements was maintained under these conditions. Thus, our data demonstrate that H3K9me3 is not sufficient for ERV silencing in endoderm-derived cells. Based on the reduced DNA methylation upon *Setdb1* deletion in XEN cells and other cell types[8,16,17] we speculate that one important role of *Setdb1*/H3K9me3 is to maintain high levels of DNA methylation. Currently, two mechanisms are being discussed: (1) H3K9me3 can be recognized by UHRF1 resulting in recruitment of DNMT1[18,19], although this view is challenged by largely maintained DNA methylation in *Uhrf1* mutants lacking the H3K9me3 binding domain[20]. (2) *Setdb1*/H3K9me3 may protect ERVs from access of TET enzymes which would remove DNA methylation[16]. The function of DNA methylation in ERV silencing is currently not fully understood on the mechanistic level. It is possible that transcription factors which could activate ERVs bind DNA in a methylation-sensitive manner[39]. DNA methylation may also help to facilitate establishment of other chromatin modifications, or recruitment of repressive chromatin binding factors. Future studies are required to better clarify the roles of H3K9me3 and DNA methylation in this context.

A surprising finding of our study was that *IAP* de-repression upon loss of *Setdb1* was limited to extraembryonic endoderm cells in vivo. We showed this by co-staining of IAP-GAG with AFP as marker for extraembryonic endoderm cells in E7.5 embryos. During this time extraembryonic endoderm-derived cells integrate into the embryonic part of definitive endoderm and, upon integration, assume very similar transcriptional and phenotypic properties[40]. As we did not use a lineage reporter to follow extraembryonic endoderm-derived cells in later embryonic stages we cannot definitely state that all IAP-GAG expressing cells are of extraembryonic origin. However, our in vitro differentiation experiments strongly support the differential mode of ERV regulation in definitive vs. extraembryonic endoderm cells. The mechanistic basis for this differential mode of regulation remains obscure. Differences in expression of chromatin regulators or different transcription factor networks are unlikely to contribute. Recently, single cell RNA-seq analyses of endoderm cells from different embryonic stages revealed an almost indistinguishable expression pattern of cells derived from embryonic or extraembryonic origin[41]. As no significant transcriptional differences appear to exist between

embryonic vs extraembryonic endoderm-derived cells in vivo, we hypothesize that the chromatin composition of ERVs established during endoderm differentiation is epigenetically maintained. More experiments are needed to determine the exact chromatin composition of ERVs in different cell types and to understand the cell type-specific recruitment of chromatin modifying factors. The mixed ontogeny of endoderm cells provides a unique opportunity to study the role and the mechanisms of epigenetic inheritance in different developmental stages and during aging in an in vivo model.

### Limitations of this study
*Setdb1* depletion in in vitro DE differentiation has minor effects on transcription of *IAPEz* elements and H3K9me3 is largely maintained. The intricacies of the DE differentiation protocol do not allow monitoring the fate of individual differentiated cells. In particular, we cannot determine how many cell divisions after *Setdb1* depletion have passed. It is, therefore, possible that even after extensive DE differentiation (day 14), cells have not undergone enough replication rounds to passively lose all *Setdb1*-dependent H3K9me3. We may therefore underestimate the role of *Setdb1* in establishing H3K9me3 in DE cells.

## Methods
### Generation of conditional Setdb1 knock-out strains
*Setdb1^(ß-gal)* mice (*Setdb1^(tm1a(EUCOMM)Wtsi)*) were crossed with actin-Flp recombinase mice[42] to remove the ß-gal cassette, resulting in the *Setdb1^flox* allele. To induce the specific deletion of *Setdb1* in endoderm cells, *Setdb1^flox* was combined with the *Sox17^(2A-iCre)* mouse line[28]. To monitor *Sox17-2A-iCre* activity, the *CAG-CAT-EGFP* reporter allele was introduced[43].

Housing of mice was performed in the BMC animal core facility which is licensed by local authorities (Az. 5.1-5682/LMU/BMC/CAM, approved on 02-12-2017 by Landratsamt München) following the regulations of German Law (TierSchG, BGBl. I S. 1206, 1313).

### Establishment of ESC lines and cell culture conditions
Blastocysts were isolated at 3.5 dpc and cultured on feeder coated 48 well plates with ESC medium containing MEK-1 inhibitor (PD098059; New England Biolabs). After 5-7 days, the inner cell mass outgrowth was trypsinized and transferred to larger feeder-coated plates. Established ESC lines were split every 2 days.

MEF and ESCs were cultured in medium based on DMEM (D6429, Sigma) containing 15% FCS (F7542 Sigma), non-essential amino acids

(M7145, Sigma), Penicillin/Streptomycin (P4333; Sigma), and 2-mercaptoethanol (Gibco, 31350-010). For ESC culture, the medium was supplemented with leukemia inhibitory factor (LIF).

## Whole mount in situ hybridization

In situ hybridization of whole-mount embryos was performed as previously described[44].

## Whole mount embryo immunostaining

Embryos were isolated in PBS+ dissection medium [PBS containing Mg$^{2+}$ and Ca$^{2+}$]. Isolated embryos were fixed for 20 min at RT in 2% PFA in PBS+ followed by permeabilizing for 10-15 min in permeabilization solution [0.1 M glycine/0.1% Triton X-100]. Embryos were transferred into blocking solution [0.1% Tween-20; 10% FCS; 0.1% BSA; 3% Rabbit, Goat or Donkey serum]. Primary antibodies were added into the blocking solution and incubated o/n at 4 °C. The following antibodies were used: Foxa2 (Abcam, ab40874, 1:1000), Sox17 (Acris/Novus, GT15094,1:1000), GFP (Aves, GFP1020, 1:1000), AFP (R&D, AF5369, 1:1000) and IAP-GAG (Cullen lab, 1:1000). The next day, embryos were kept at RT for 2 hours. After 3 washes with PBST, embryos were incubated with secondary antibodies Donkey anti rabbit Alexa488 (Jackson Immuno Research, 711-545-152, 1:800), Donkey anti goat Alexa 555 (Invitrogen, A-21432, 1:1000), Donkey anti Chicken IgY Alexa488 (Jackson Immuno Research, 703-545-155, 1:800), Donkey anti mouse Cy3 (Jackson Immuno Research, 715-165-150, 1:800), Donkey anti rabbit Alexa647 (Jackson Immuno Research, 711-605-152, 1:500) for 3 hours at RT, followed by three washes. Embryos were then embedded in antifade medium (Invitrogen, P36930) for microscopy analysis.

## β-galactosidase staining and histology

Embryos were fixed with 4% paraformaldehyde/PBS at 4 °C [E7.5/5 min; E8.5/10 min; E9.5/20 min]. After washes with LacZ rinse solution [2 mM MgCl2; 0.02% NP-40; 0.01% sodium deoxycholate in PBS], embryos were stained with X-gal staining solution [1 mg/ml dimethylformamide; 5 mM potassium ferricyanide; 5 mM potassium ferrocyanide in LacZ rinse solution] o/n at 37 °C.

For histological sections, embryos were fixed overnight in 4% formaldehyde and embedded in paraffin. The embedded embryos were sectioned and stained with Hematoxylin and Eosin.

## In vitro endoderm differentiation

For in vitro differentiation towards definitive endoderm 0.1 Mio ESCs were seeded on FCS coated 6-well plates directly into endoderm differentiation medium (EDM) [500 ml Advanced DMEM/F-12 (1x) (Gibco/LifeTechnologies; 12634-10- 500 ml), 500 ml Advanced RPMI 1640 (1x) (Gibco/LifeTechnologies; 12633-012- 500 ml), 22 ml GlutaMAXTM – I CTSTM (Gibco/LifeTechnologies; 12860-01- 100 ml), 200 µl AlbuMAX 100 mg/ml (Gibco/LifeTechnologies; 11021-029 100 g, 22 ml HEPES 1 M (Gibco/LifeTechnologies; 15630-056- 100 ml), 70 µl Cytidine 150 mg/ml (SIGMA; C4654-5G), 0,9 ml ß-Mercaptoethanol 50 mM (Gibco/LifeTechnologies; 31350-10- 20 ml), 12 ml Pen/Strep (10000U/ml) (Gibco/LifeTechnologies; 10378016 – 100 ml), 1 ml Insulin-Transferin-Selenium Ethanolamine (Gibco/LifeTechnologies; 51500-056- 10 ml)], supplemented with 2 ng/ml of murine Wnt3a (1324 WN-CF, R&D systems) and 10 ng/ml of Activin A (338-AC, R&D systems). Cells were collected on day 7 for FACS isolation.

For in vitro differentiation towards extraembryonic endoderm, 0.2 million ESCs were seeded on gelatine coated 6-well plates directly in ESC medium and then were transduced with a lentiviral *Gata6* overexpression construct (#1582 pLenti6/EF1a-GATA6-IRES-Puro) on the next day. Two days after transduction, *Gata6* expressing cells were selected with 1 µg/ml puromycin. Five days after transduction, ESC medium was replaced with XEN medium [Advanced RPMI 1640 (1x) (Gibco/LifeTechnologies; 12633-012- 500 ml), supplemented with 15%

FCS, 0.1 mM β-mercaptoethanol and 1% penicillin-streptomycin]. Cells were collected on day 7 for FACS isolation.

For the extended DE differentiation time-course cells were harvested at day 8 of differentiation. DE cells were detached with StemPro™ Accutase™ Cell Dissociation Reagent (Thermofisher) by incubation for 3 minutes at 37 °C. For later time-points cells were re-seeded onto Biotechne Cultrex Stem Cell Qualified Reduced Growth Factor BME coated dishes and harvested at day 10, 12 and 14.

## Cell cycle analysis

Cell cycle analysis of DE cells was performed using Hoechst 33342 Ready-Flow Reagent (Thermo Fisher Scientific) at 1 drop per 0.5 ml of cell suspension, incubated at 37 °C for 10 min. Cells were analyzed on FACSAriaFusion SORP or LSRFortessa SORP (both BD Biosciences) equipped with a UV laser (355 nm) for optimal excitation of Hoechst 33342. Dead cells were excluded using SYTOX™ Red Dead Cell Stain, for 633 or 635 nm excitation (ThermoFisher) in a 1:1000 dilution. Cell cycle profiles were analyzed with FlowJo v10.8.1 (BD). Doublets were excluded from analysis based on SSC-H versus -W and Hoechst-W versus -H plots.

## Immunofluorescence microscopy

Cells were carefully washed once with PBS. Fixation was carried out with 3.7% formaldehyde (Carl Roth) in PBS for 10 min at RT. Cells were then permeabilized with 3 mM sodium citrate tribasic dehydrate (Merck), 0.1% v/v Triton X-100. Permeabilized cells were washed twice with PBS and twice in washing solution [PBS, 0.1% v/v Tween 20, 0.2% w/v BSA] for 5 min. Cells were blocked with blocking solution [PBS, 0.1 % v/v Tween 20, 2.5% w/v BSA] for 30 min and incubated overnight at 4 °C with primary antibodies in blocking solution. The following antibodies were used: GFP (Aves, GFP1020, 1:1000), IAP-GAG (Cullen lab, 1:1000). Cells were washed three times in washing solution for 10 min before incubation with secondary antibodies Donkey anti Chicken IgY Alexa488 (Jackson Immuno Research, 703-545-155, 1:800) and Donkey anti rabbit Alexa 555 (Invitrogen, A-31572, 1:1000) in blocking solution containing 10% normal goat serum (Dianova-Jackson Immuno Research) at RT for 1 h. After washing three times in PBS, 0.1% Tween 20 for 10 min, cells were embedded with Vectashield/DAPI (Vector Laboratories) on standard microscope slides (Carl Roth). The immunofluorescence staining was examined with Axiovert 200 M inverted microscope for transmitted light and epifluorescence (Carl Zeiss Microscopy) with the help of the AxioVision Special Edition Software (Carl Zeiss Microscopy).

## Fluorescence activated cell sorting

Cells were resuspended in PBS/0.2% FCS before FACS collection. Cells from embryos were directly sorted into lysis buffer (Thermo Fisher, KIT0204) followed by RNA extraction. Cells from in vitro culture were sorted into PBS/0.2% FCS. FACS was performed using a FACS Aria instrument (BD Biosciences). Data were analysed using FlowJo software.

## RT-qPCR analyses

Total RNA from three independent biological replicates of sorted cells was isolated using the RNA Clean & Concentrator kit (Zymo Research) including digestion of remaining genomic DNA according to producer´s guidelines. qPCR was carried out with the Fast SYBR Green Master Mix (Applied Biosystems) in a LightCycler480 (Roche) according to the Fast SYBR Green Master Mix-protocol. Primers were evaluated for generating a single PCR product and for linear amplification in a wide range of DNA template dilutions. Every PCR-reaction was performed in a total volume of 10 µl in duplicates, triplicates or quadruplicates in a 384-well plate (Sarstedt). Two independent control genes (*Gapdh* and *HPRT*) were used as reference genes for qRT-PCR experiments and geometric mean of reference Ct values was used as normalization[45].

For qRT-PCR of repetitive regions like *IAP* elements, negative control samples that were not treated with reverse transcriptase were used to control for genomic DNA background. Ct-values were generated by the LightCycler480-Software (Roche) using the 2nd derivative max function and fold changes were calculated using the $2^{-\Delta\Delta Ct}$ method.

## Western blot

Whole cell proteins extracts were prepared by resuspending 1 million cells in 40 µl of freshly prepared lysis buffer containing 50 mM Tris/HCl pH 7.5, 2% w/v SDS, 1% v/v Triton X-100, 1 mM PMSF, 0.5x Roche Complete Protease Inhibitor Cocktail. Samples were vortexed for 10 s at max speed and boiled for 10 min at 95 °C. After incubation with 1 µl of Benzonase/ 2.5 mM $MgCl_2$ at 37 °C for 15 min, protein extracts were mixed with 12 µl 4x sample buffer (Roth) and boiled again for 5 min at 95 °C. The boiled protein extracts were separated through SERVAGel TG PRiME 4-12 % precast SDS Page (SERVA Electrophoresis) in running buffer 25 mM Tris, 200 mM glycine, 1% (m/v) SDS at RT for 1 h and 25 mA per gel. Gels were blotted onto methanol activated PVDF membranes in a wet-blotting chamber (Bio-Rad Laboratories) containing blotting buffer 50 mM Tris, 40 mM glycine, 10% v/v methanol, 5 µM SDS for 1.5 h at 4 °C. Membranes were incubated in blocking buffer 1x PBS, 2.5% w/v BSA and 2.5 % w/v milk at RT for 1 h under mild agitation. Blocked membranes were incubated with primary Ab in blocking buffer at 4 °C for 16 h. The antibodies used were Setdb1 (Santa Cruz, sc66884-X, 1:250) and α-Tubulin (Sigma, T5168, 1:1000). Membranes were washed 3 times with PBST buffer 1x PBS, 0.1% v/v Tween 20 for 20 min. Incubation with secondary Ab 680RD Goat anti-Mouse (LI-COR, 926-68070, 1:3000) and 800CW Goat anti-Rabbit (LI-COR, 926-32211, 1:3000) was done in blocking buffer at RT for 1.5 h. The probed membranes were washed 3 times in PBST for 20 min. Based on the detection method, Immobilon Western Chemiluminescent HRP Substrate (Merck Millipore) was used for ECL method and IRDye 800CW Secondary Antibodies for LI-Cor method. Chemoluminescence was detected in by ChemiDoc MP Imaging System with the Image Lab Software using ECL Western blot detection reagent (Amersham Biosciences) or by Li-Cor Odyssey Imaging System with the Image studio software.

## RNAseq analysis

The Agilent 2100 Bioanalyzer was used to assess RNA quality and only high-quality RNA samples (RIN > 8) were further processed for cDNA synthesis using SMART-Seq v4 Ultra Low Input RNA Kit (Clontech cat. 634888) according to the manufacturer´s instruction. cDNA was fragmented to an average size of 200-500 bp in a Covaris S220 device (5 min; 4 °C; PP 175; DF 10; CB 200). Fragmented cDNA was used as input for library preparation using MicroPlex Library Preparation Kit v2 (Diagenode, cat. C05010012) and processed according to the manufacturer´s instruction. RNA samples from Dnmt1 ko cells were Ribo-depleted using the NEBNext rRNA Depletion Kit (Human/Mouse/Rat) (NEB #E6310) and, RNAseq libraries were generated using the NEBNext Ultra II Directional RNA Library Prep Kit for Illumina (NEB #E7760) according to the manufacturer's instructions. All libraries were quality controlled by Qubit and Agilent DNA Bioanalyzer analysis. Deep sequencing was performed on HiSeq 1500 system according to the standard Illumina protocol for 50 bp single end reads.

## ChIP-seq of histone modifications

0.5 million FACS-sorted cross-linked cells (1% formaldehyde, 10 min RT) were lysed in 100 µl Buffer-B-0.5 (50 mM Tris-HCl, pH 8.0, 10 mM EDTA, 0.5% SDS, 1x protease inhibitors -Roche) and sonicated in a microtube (Covaris; 520045) using a Covaris S220 device until most of the DNA fragments were between 200-500 base pairs long (settings: temperature 4 °C, duty cycle 2%, peak incident power 105 Watts, cycles per burst 200). After shearing, lysates were centrifuged for 10 min, 4 °C, 12000 g and supernatant diluted with 400 µl of Buffer-A (10 mM Tris-HCl pH 7.5, 1 mM EDTA, 0.5 mM EGTA, 1% Triton X-100, 0.1% SDS, 0.1% Na-deoxycholate, 140 mM NaCl, 1x protease inhibitors-Roche). 150 µl of sonicated chromatin was then incubated 4 h at 4 °C on a rotating wheel with 3 µg of H3K9me3 antibody (Active Motif; 39161) conjugated to 10 µl of magnetic beads. Beads were washed four times with Buffer-A (10 mM Tris-HCl pH 7.5, 1 mM EDTA, 0.5 mM EGTA, 1% Triton X-100, 0.1% SDS, 0.1% Na-deoxycholate, 140 mM NaCl, 1x protease inhibitors - Roche) and once with Buffer-C (10 mM Tris-HCl pH 8.0, 10 mM EDTA). Beads were re-suspended in 100 µl elution buffer (50 mM Tris-HCl, pH 8.0, 10 mM EDTA, 1% SDS) and incubated 20 min at 65 °C. Supernatant was transferred to a new tube. Crosslink reversal of immunoprecipitated DNA was carried out overnight at 65 °C. Then 100 µl TE (10 mM Tris-HCl pH 8.0, 1 mM EDTA) was added, RNA was degraded by 4 µl RNase A (10 mg/ml) for 1 hour at 37 °C and proteins were digested with 4 µl Proteinase K (10 mg/ml) at 55 °C for 2 hours. Finally, DNA was isolated by phenol:chloroform:isoamyl alcohol purification followed by ethanol precipitation. Purified DNA was used as input for library preparation using MicroPlex Library Preparation Kit v2 (Diagenode, cat. C05010012) and processed according to the manufacturer´s instruction. Libraries were quality controlled by Qubit and Agilent DNA Bioanalyzer analysis. Deep sequencing was performed on HiSeq 1500 system according to the standard Illumina protocol for 50 bp single-end reads.

For the extended DE differentiation time-course cells (day 12 and day 14) ChIPseq was was done using the following protocol. Briefly, 50.000 FACS-sorted cross-linked cells (1% formaldehyde, 10 min RT) were lysed in 100 ul Buffer-B-0.3 (50 mM Tris-HCl, pH 8.0, 10 mM EDTA, 0,3%SDS, 1x protease inhibitors -Roche) and and sonicated in a microtube (Covaris; 520045) using a Covaris S220 device until most of the DNA fragments were between 200-500 base pairs long (settings: temperature 4 °C, duty cycle 2%, peak incident power 105 Watts, cycles per burst 200). After shearing, lysates were diluted with 1 volume of Dilution Buffer (1 mM EGTA 300 mM NaCl, 2% Triton x-100, 0.2% sodium deoxycholate, 1x protease inhibitors-Roche). Sonicated chromatin) was then incubated 4 h at 4 °C on a rotating wheel with 1 ug of H3K9me3 (Diagenode C15410193) antibody conjugated to 10 µl of Protein-G Dynabeads (Thermofisher). Beads were washed four times with Buffer-A (10 mM Tris-HCl, pH 7.5, 1 mM EDTA, 0.5 mM EGTA,1% Triton X-100, 0.1% SDS, 0.1% Na-deoxycholate, 140 mM NaCl, 1x protease inhibitors) and once with Buffer-C (10 mM Tris-HCl, pH 8.0, 10 mM EDTA). Beads were then incubated with 70 µl elution buffer (0.5% SDS, 300 mM NaCl, 5 mM EDTA, 10 mM Tris HCl pH 8.0) containing 2 µl of Proteinase K (20 mg/ml) for 1 hour at 55 °C and 8 hours at 65 °C to revert formaldehyde crosslinking, and supernatant was transferred to a new tube. Another 30 µl of elution buffer was added to the beads for 1 minute and eluates were combined and incubated with another 1 µl of Proteinase K for 1 h at 55 °C. Finally, DNA was purified with SPRI AMPure XP beads (Beckman Coulter) (sample-to-beads ratio 1:2). Purified DNA was used as input for library preparation with Thruplex DNA-seq kit (Takara, cat. R400674) and processed according to the manufacturer´s instruction. Libraries were quality controlled by Qubit and Agilent DNA Bioanalyzer analysis. Deep sequencing was performed on Illumina NextSeq device.

## Oxidative bisulfite analysis

Genomic DNA was prepared using the DNEasy Blood and Tissue Kit (Qiagen) and subjected to bisulfite conversion using the EpiTect Bisulfite Kit (Qiagen) according to the manufacturer's protocol. Jumpstart Taq polymerase (Sigma Aldrich) was used to amplify the IAP GAG region, IAP LTR region and a 200 bp region of LINE-1. PCR primers for bisulfite-converted DNA were modified by adding Illumina adaptors for library preparation based on previous studies[15,46,47]. The gel-purified amplicons were indexed with index primers/universal PCR

primers and Illumina P5/P7 primers. Before amplification, the DNA was purified with SPRI AMPure XP beads (sample-to-beads ratio 1:0.8). Libraries were checked for quality control and correct fragment length on a Bioanalyzer 2100 (Agilent) and concentrations were determined with Qubit dsDNA HS Assay Kit (Life Technologies). Sequencing was carried out on a MiSeq sequencer (2 × 300 bp and 2 × 250 bp paired end) with v3 chemistry (Illumina).

### Intracellular Staining for IAP-GAG
ESCs and endoderm differentiated cells were resuspended in 500 μl PBS containing 2 μl zombie aqua (Biolegend, cat no.423101). Fixation/Permeabilization was performed in 1 ml of Foxp3 fixation/permeabilization buffer. After 30 min incubation at 4 °C in the dark, samples were washed with 2 ml of 1x permeabilization buffer and centrifuged at 400 g at RT for 5 min. The pellet was resuspended in 100 μl of 1x permeabilization buffer after a second wash and incubated with primary antibodies IAP-GAG (Cullen lab, 1:1000) and Sox17 (Acris/Novus, GT15094, 1:1000) for at least 30 minutes at room temperature in the dark. After washes with 2 ml of 1x permeabilization buffer, the pellet was resuspended in 100 μl of 1x permeabilization buffer and incubated with secondary antibodies anti-rabbit (Jackson; 711605152, 1:800) and anti-goat (Invitrogen; A21432, 1:1000) at room temperature for 60 min in the dark. After two washes with 2 ml of 1x permeabilization buffer, cells were resuspended in 300 μl of FACS buffer. ESCs and differentiated cells were then analyzed by flow cytometry using FACS Aria instrument (BD Biosciences). Data were further processed using the FlowJo v10 Software. FITC-A channel (Sox17) was used to distinguish endoderm differentiated from undifferentiated cells.

### Reagents
Cell lines, antibodies, primers and plasmids are listed in the Supplementary Information.

### Bioinfomatic analysis
**RNA-seq.** Single end reads were aligned to the mouse genome version mm10 using STAR[48] with default options "--runThreadN 32 --quantMode TranscriptomeSAM GeneCounts --outSAMtype BAM SortedByCoordinate". Read counts for all genes and repeats were normalized using DESeq2[49]. Significantly changed genes were determined through pairwise comparisons using the DESeq2 results function (adjusted $p$ value < 0.01). The expression levels of different repeat classes was assessed using Homer through analyzeRepeats.pl with the repeats function. The repeat definitions were loaded from UCSC. Significantly changed ERV families were determined through pairwise comparisons using the DESeq2 results function (log2 fold change threshold = 1, adjusted $p$ value < 0.01). PCA analyses were done using the plotPCA function of the DESeq2 package. Bargraphs showing expression data for selected genes were plotted using ggplot2 with RSEM-normalized data (TPM = Transcript Per Million). Heatmap with differentially expressed ERV families was plotted with pheatmap using rlog-normalized expression values.

**ChIP-seq.** ChIP-seq single end reads were aligned to the mouse genome mm10 using Bowtie with options "-q -n 2 --best --chunkmbs 2000 -p 32 -S". The H3K9me3 enrichment of different repeat classes was assessed using Homer through analyzeRepeats.pl with the repeats function. The repeat definitions were loaded from UCSC. Correlation of expression and H3K9me3 changes for ERVs were plotted by log2-foldchange of H3K9me3 enrichment over input versus log2foldchange of expression. Cumulative read coverage across IAP elements was calculated using coverageBed and normalized to the library size. Coverage profiles were plotted using ggplot2.

H3K9me3 domains were identified using chromstaR[50] with options binsize = 5000, stepsize = 1000 in mode "separate". Differential

H3K9me3 peaks were detected with chromstaR using options binsize = 1000, stepsize = 500 in mode "differential".

**Bisulfite sequencing.** Bisulfite sequencing paired end reads were aligned using CLC Genomics Workbench. Methylation analysis of sequencing data was performed using QUMA: quantification tool for methylation analysis[51].

### Reporting summary
Further information on research design is available in the Nature Research Reporting Summary linked to this article.

## Data availability
The data that support this study are available from the corresponding author upon reasonable request. NGS data (Supplementary Data 13) were deposited on NCBI GEO database with accession number GSE139128. Source data are provided with this paper.

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

## Acknowledgements

We deeply thank Brian Cullen for providing his last aliquots of the IAP-GAG antibody. We wish to thank the help of BMC core facilities for FACS and Bioinformatics for providing equipment, services, expertise, and assistance with data analysis. We are further thankful to the LAFUGA Sequencing facility and the Genomics Service Unit of the LMU. Funded by the Deutsche Forschungsgemeinschaft (DFG, German Research Foundation), Project-ID 213249687 – SFB 1064 to G.S. and H.Le. and Projekt-ID 329628492 – SFB 1321 TP13 to G.S.

## Author contributions

G.S., H.Li., H.Le., Z.W., R.F. contributed to concepts and approaches (designed the experimental approach) (conceived and designed the project); Z.W., R.F., A.R., F.M.C., A.N., S.S., I.S., I.D., E.U., T.A., L.R. performed the experiments; Z.W. performed the bioinformatic analysis with the help of G.S.; G.S. and Z.W. wrote the manuscript.

## Funding

## Competing interests

The authors declare no competing interests.
