## [Peer Review File · Nature Communications]

Reviewers' comments:

Reviewer #1 (Remarks to the Author):

Wang et al (MS ID#: NCOMMS-20-02667)

In this study, the authors described that two different types of endoderm-lineage cells, thus visceral endoderm (VE) and definitive endoderm (DE), use different mode for endogenous retrovirus (ERV) silencing. In VE progenitor cells, histone H3 lysine 9 (H3K9) methyltransferase SETDB1 was critical for silencing of Intracisternal A-particle (IAP) elements, but it was dispensable for DE cells. On the other hand, DNA methyltransferase Dnmt1 was essential for IAP silencing in both VE and DE cells. Importantly, Dnmt1 knock-out did not induce H3K9me3 reduction on such ERV elements. From these results, the authors suggest that DNA methylation has a dominant role for ERV silencing in endoderm cells over H3K9me3.

It is interesting that impact of SETDB1 depletion for IAP silencing is different between VE and DE cells, even both in endoderm-lineage cells. Experiment using in vitro extraembryonic endoderm (XEN) and DE differentiated cells from same Embryonic stem cells (ESCs) also reproduce this difference, which is also nice. However, it is already shown that DNA methylation is essential but SETDB1 and SETDB1-mediated H3K9me3 is dispensable for IAP silencing in other differentiated cells (PMCID:PMC5923290).

Furthermore, H3K9me3 is maintained in Setdb1 KO DE cells, still possible that H3K9me3 play additive role for IAP silencing in DE cells other than DNA-methylation pathway. In ESCs, it is suggested that both H3K9me3 and DNA methylation contribute to IAP silencing (PMCID:PMC3857791). So, some of data are potentially interesting such as 1) Setdb1 KO in XEN cells induces loss of H3K9me3 (entire) and reduction of DNA methylation (GAG region) on IAP and 2) Setdb1 KO in DE cells does not diminish the H3K9me3 levels on IAP, but current study does not provide further mechanistic insights of these differences for IAP silencing. Also, part of IAP elements, mainly IAPez is only depressed in Dnmt1-KO cells, thus "IAP specific silencing" but not "ERV silencing" is appropriate for discussion. Following is more specific comments,

Major comments

1. The authors mentioned that SETDB1 is critical to maintain H3K9me3 and DNA methylation on ERVs in XEN, but not in DE cells. However, this finding is specific to IAP families not to all ERVs. For example, even in Setdb1 KO DE cells, ERVs other than IAPs are derepressed (Fig. 3D right) and level of H3K9me3 on some of ERVs are also diminished (RLTR5_Mm, MURVY-int and more below -1 level in Fig. 4A right). On the other hand, in Dnmt1 KO, derepressed ERVs are mostly IAP families in both XEN and DE cells. Thus, other ERV families are still silenced by Setdb1 or H3K9me3. This is consistent with the published results (PMCID:PMC5923290). So, the authors final argument is only IAP specific and dominant role of Dnmt1 on IAP silencing itself is not novel.

2. The authors said that in discussion, “We could not detect IAP-GAG expression in Suv39h double ko definitive endoderm cells (data not shown), suggesting a similar synergism between Setdb1 and Suv39h enzymes in early definitive endoderm.” . First of all, the authors should include this data instead of “data not shown” . 2nd, to prove this possibility of synergism, the authors should try Setdb1 KO or KD using siRNA on the top of Suv39h dKO DE to test if H3K9me3 and DNA methylation is affected or not, and also examine whether IAP is depressed or not.

3. Sup Fig 3A, B

Related to the issue of 2,

The authors argued that Setdb1 KO in DE cells does not diminish the H3K9me3 levels on IAP. However, % of GFP positive population between the in vitro differentiation culture for XEN- and DE-lineaged cells are different (88.4% for XEN and 44.4% for DE) and Setdb1 mRNA is still detected in DE cells but not XEN cells, suggesting that timing of Setdb1 deletion might be different between two populations. If Setdb1 depletion occurs later timing in DE cells than XEN cells, this comparison is not appropriate. Even though SETDB1 is similarly depleted in XEN and DE cells at the analyzed time point (Fig. S3C), the authors should check timing of SETDB1 depletion in this in vitro differentiation culture and compare at more appropriate timing if timing of SETDB1 depletion is different between two culture conditions.

Minor comments

1. The authors generalized derepression of IAP is restricted to VE and does not occur in DE using mouse model and in vitro differentiated XEN and DE cells derived from Setdb1^{END} ESCs. The authors clearly showed that the difference of IAP derepression between Setdb1 KO XEN and DE cells. However, in vivo part it is not clear. For example, why is the red signal increased even in extraembryonic part in the GAG staining in Fig. 2C ? Also, in Fig. 2D, the authors indicated AFP positive cells by white arrowheads. However, the GAG staining signal is increased in the entire region of Setdb1^{END} embryo.

2. In Fig. 4, Is it possible to do SETDB1 ChIP experiments to examine whether SETDB1 is localized on IAP locus both in XEN cells and DE cells? This exp also clarify whether SETDB1 is not involved in H3K9me3 on IAP in DE cells.

3. Fig. 5

DNA methylation status on ERVs should be shown in control and Dnmt1 KO ES, XEN and DE cells

3. Supplementary Fig. 5

Figures A-C are missing

Reviewer #2 (Remarks to the Author):

The authors demonstrated that conditional knock-out of the H3K9me3 HMTase Setdb1 in mouse embryonic endoderm results in ERV de-repression in visceral endoderm (VE) descendants but not in definitive endoderm (DE), suggesting differential regulatory mechanisms in endoderm cells depending on their developmental origin. Deletion of Setdb1 resulted in loss of H3K9me3 and reduced DNA methylation of IAP elements specifically in VE progenitors. In contrast, Dnmt1 knock-out resulted in ERV de-repression in both visceral and definitive endoderm cells without affecting H3K9me3 marks. Based on these findings, the authors concluded that Setdb1-mediated H3K9me3 is not sufficient for ERV silencing, but rather critical for maintaining high DNA methylation.

Although the experimental data are solid, differential dependency of ERV silencing on Setdb1-mediated H3K9me3 in somatic cells has already been reported by other groups. Several developing organs show ERV derepression in Setdb1 KO mice, but somatic cells in adults are rather resistant to Setdb1 deletion (Tan et al., *Development* 2012, Koide et al., *Blood* 2016; Kato et al., *Nat Commun* 2018). The finding that Dnmt1 knock-out induces ERV de-repression in endoderm cells without affecting H3K9me3 marks is interesting. This finding contrasts with the ERV silencing in ESCs, and highlights the essential role for DNA methylation in somatic cells. As the authors proposed, Setdb1-mediated H3K9me3 may function in maintaining high DNA methylation in somatic cells. In this regard, the underlying molecular mechanism is quite important. How does Setdb1-mediated H3K9me3 protect the DNA methylation levels? Additional experimental data which provides the functional link between Setdb1-mediated H3K9me3 and DNA methylation would improve the impact of this study. Other concerns are as follows.

1. Two H3K9me3 HMTases, Setdb1 and Suv39h, are assumed to function differentially as well as redundantly in ERV silencing. To understand their roles in ERV silencing, ChIP-seq of Setdb1 and Suv39h or profiling of ERV expression in Suv39h KO in comparison with Setdb1 KO would be informative.
2. Please show DNA methylation levels in Dnmt1 KO XEN and DE cells. Are they really low in Dnmt1 KO in endodermal cells?
3. Why is Nepn, an important regulator of endoderm development downregulated in Setdb1^{END} embryos?

Reviewer #3 (Remarks to the Author):

H3K9me3 is a mark of constitutive heterochromatin, and is tightly correlated with high DNA methylation. In mammals, several H3K9 methyltransferases exist, and it is not fully resolved what their specific and redundant activities are. Moreover, it is not clear what the epistatic relationship is between the histone mark and DNA methylation-based regulation. In mice, particularly mouse embryonic stem cells, the H3K9 methyltransferase SETDB1 is associated with silent endogenous retroviruses (ERVs), markedly IAPs. Given cell-type specific differences of ERV regulation, here Wang et al set out to explore the role of SETDB1 in both in vivo and in vitro models of endoderm differentiation. Using an endoderm-specific inducible knockout system, they made the interesting finding that SETDB1 appears to be important for regulating ERVs in extra embryonic endoderm (akin to its role in ESCs), and less so in definitive endoderm. The authors expounded in this finding by employing an in vitro endoderm differentiation system, consistently showing a requirement for SETDB1 in the extra embryonic (XEN) rather than embryonic (DE) cell types. Finally, the authors attempt to untangle the relationship between H3K9 and DNA methylation using a Dnmt1 mutant. Whereas DNA methylation plays an important role in repression of ERVs in both XEN and DE cells, surprisingly H3K9me3 is unaffected at the IAPez family of ERVs. This indicates that DNA methylation is the more downstream effector of silencing of ERVs, and is an intriguing result. I should say that I remain partially skeptical about this result (see below).

Overall, the paper is concise and clearly expounded. Moreover, I found the combination of elegant mouse genetics with an in vitro system to be a strong validation of their results. However, I did find the paper to be somewhat deficient in some areas. While the finding that SETDB1 has cell type-specific effects is interesting and worth publishing, per se, there is very little experimental digging into the how (see major point) or the why. What is the biological underpinning for these changes during development, and why would there be such a stark difference between embryonic and extra embryonic tissues? As it stands, the paper is missing these key elements to push it into a journal such as Nature Communications.

Major Points

- The authors make an important finding that SETDB1 only has an importance in extra-embryonic endoderm tissues, however they never explore the genetic interactions between the various HMTs. I understand that performing further genetic tests in the complicated in vivo system would be very time consuming, however they have implemented an ESC system where these experiments should be relatively straightforward to achieve. I suggest comparing Setdb1, suvr39 dKO, and the triple knockout cells in parallel. A conditional knockout (or degron) system could be utilized in case the triple knockout is not viable. As the authors mentioned, HMTase tKO data was published recently (Nicetto et al, 2019). However, I believe the authors could truly expand upon their findings by exploring the effects on histone methylation, DNA methylation, and expression in combinatorial mutant lines. Partially related, I would have liked to have seen more analyses presented from the ChIP-Seq data. Were there any regions where H3K9me3 was impacted in the Setdb1 mutant line? The authors only present data from retrotransposons, and do not discuss other genomic compartments in the text. This could be more flushed out with increased results in other genetic backgrounds.

- The authors make the claim that in the Dnmt1 knockout cells, expression of ERVs is higher, but H3K9me3 at IAPez remains unchanged, seen by both ChIP-qPCR and -seq. However, given that these are repetitive elements, how can one conclude that the increased expression is not due to a small number of highly expressed loci that are NOT marked for H3K9me3? These would be simply missed from the analysis. I understand this is a hard point to address without using long read sequencing technology, but it should at least be discussed as a possibility before affirming the genetic relationship. I am not aware of any biological situations where H3K9me3-marked loci are associated with such high expression, even in the case of heterochromatic RNAs. Of course, it being biology, you never know.

Minor Points

- With regards to embryology, I am not an expert and it would have been useful to have a simple schematic for endoderm differentiation in the supplement, so those (such as me) would not need to refer to google.

- Abstract: A tad confusing to describe modifications on ERVs as “established” by the “maintenance” DNA methyltransferase. Perhaps use another verb, such as “deposited”

- Line 132: Nepn downregulation is likely a secondary effect, I presume. It’s striking how much stronger its downregulation is compared to any other gene. Could the authors speculate? Does it serve as a sensor for malforming endoderm tissue?

- Line 138: Not all LINE families are unchanged. It appears one is significantly unregulated (L1Md_T|LINE|L1). This should be at least acknowledged
- It is never clear how many embryos are being assessed for each experiment. For example, in Figure 2D, is the statement about VE and GAG expression being concluded from only one embryo? This information should at least be added to the figure legends.
- Line 211 and Figure S4B: Given that LINE K9me3 methylation is not impacted in Setdb1 mutant XEN cells, I'm surprised that DNA methylation and expression are. The authors never discuss this curiosity. What is the explanation? Perhaps it is because Setdb1 is required for recruiting de novo methylation through DNMT3A and/or B during differentiation, which clearly must occur on these elements?
- In the PDF, Figure panels S5A-C appear to be missing, which are important control experiments.
- I would place Figure S5E in the primary, and put the CHIP qPCR in the supplemental. I found the CHIP-Seq data the more compelling result.

Reviewers' comments:

Reviewer #1 (Remarks to the Author):

Wang et al (MS ID#: NCOMMS-20-02667)

In this study, the authors described that two different types of endoderm-lineage cells, thus visceral endoderm (VE) and definitive endoderm (DE), use different mode for endogenous retrovirus (ERV) silencing. In VE progenitor cells, histone H3 lysine 9 (H3K9) methyltransferase SETDB1 was critical for silencing of Intracisternal A-particle (IAP) elements, but it was dispensable for DE cells. On the other hand, DNA methyltransferase Dnmt1 was essential for IAP silencing in both VE and DE cells. Importantly, Dnmt1 knock-out did not induce H3K9me3 reduction on such ERV elements. From these results, the authors suggest that DNA methylation has a dominant role for ERV silencing in endoderm cells over H3K9me3.

It is interesting that impact of SETDB1 depletion for IAP silencing is different between VE and DE cells, even both in endoderm-lineage cells. Experiment using in vitro extraembryonic endoderm (XEN) and DE differentiated cells from same Embryonic stem cells (ESCs) also reproduce this difference, which is also nice. However, it is already shown that DNA methylation is essential but SETDB1 and SETDB1-mediated H3K9me3 is dispensable for IAP silencing in other differentiated cells (PMCID:PMC5923290). Furthermore, H3K9me3 is maintained in Setdb1 KO DE cells, still possible that H3K9me3 play additive role for IAP silencing in DE cells other than DNA-methylation pathway. In ESCs, it is suggested that both H3K9me3 and DNA methylation contribute to IAP silencing (PMCID:PMC3857791). So, some of data are potentially interesting such as 1) Setdb1 KO in XEN cells induces loss of H3K9me3 (entire) and reduction of DNA methylation (GAG region) on IAP and 2) Setdb1 KO in DE cells does not diminish

the H3K9me3 levels on IAP, but current study does not provide further mechanistic insights of these differences for IAP silencing. Also, part of IAP elements, mainly IAPez is only depressed in Dnmt1-KO cells, thus "IAP specific silencing" but not "ERV silencing" is appropriate for discussion. Following is more specific comments,

We appreciate that the reviewer finds our study interesting. Indeed, Kato et al., 2018 have shown that DNA methylation is important for IAP silencing in somatic cells and that Setdb1 is dispensable for IAP silencing in some somatic cell types. However, this study did not investigate the relation between Setdb1 or Dnmt1 loss on H3K9me3, DNA methylation and IAP transcription in the same cell type. To our knowledge there is no published report of IAP de-repression while fully maintaining H3K9me3. In fact, also reviewer #3 found it quite surprising to see IAP derepression with maintained H3K9me3.

Based on the early findings of abundant H3K9me3 in pericentric heterochromatin by Suv39h enzymes and later, the demonstration that Setdb1 targets H3K9me3 to ERV families, H3K9me3 evolved as the central dogma for "heterochromatin" and "silencing". Our data clearly demonstrate that in Dnmt1 ko XEN and DE cells, H3K9me3 is maintained while IAPs are de-repressed. So, it clearly is not mediating the actual silencing process. Interestingly, these data agree with a recent study by the Torres-Padilla lab, which describe non-silencing roles of H3K9me3 in early embryos (PMID: 32601371). Obviously, we have not deciphered the full IAP silencing mechanism, which is studied for more than 10 years and involves numerous chromatin regulators that interplay with (still unknown) transcriptional activators. However, we think it is important to communicate our findings to the broader community to stimulate questions regarding the true function of H3K9me3 in the context of transcriptional silencing.

Major comments

1. The authors mentioned that SETDB1 is critical to maintain H3K9me3 and DNA methylation on ERVs in XEN, but not in DE cells. However, this finding is specific to IAP families not to all ERVs. For example, even in Setdb1 KO DE cells, ERVs other than IAPs are derepressed (Fig. 3D right) and level of H3K9me3 on some of ERVs are also diminished (RLTR5_Mm, MURVY-int and more below -1 level in Fig. 4A right). On the other hand, in Dnmt1 KO, derepressed ERVs are mostly IAP families in both XEN and DE cells. Thus, other ERV families are still silenced by Setdb1 or H3K9me3. This is consistent with the published results (PMCID:PMC5923290). So, the authors final argument is only IAP specific and dominant role of Dnmt1 on IAP silencing itself is not novel.

The reviewer points out that our study mainly focuses on IAP regulation. To make this clearer we decided to change the title of our manuscript to “Dominant role of DNA methylation over H3K9me3 for IAP silencing in endoderm”. The reason for the focus on IAP regulation is the fact that it is already known that both DNA methylation and H3K9me3 are important in different tissues or cell types (for example in PMCID:PMC5923290 as stated by the reviewer), but in none of these cases a consistent analysis of the interdependence between these “silencing marks” has been performed. Either the study was focused on H3K9me3 or it was focused on DNA methylation. Therefore, our finding that DNA methylation loss is compatible with H3K9me3 maintenance and transcriptional de-repression of IAPs is clearly novel.

2. The authors said that in discussion, “We could not detect IAP-GAG expression in Suv39h double ko definitive endoderm cells (data not shown), suggesting a similar synergism between Setdb1 and Suv39h enzymes in early definitive endoderm.”. First of all, the authors should include this data instead of “data not shown”. 2nd, to prove this possibility of synergism, the authors should try Setdb1 KO or KD using siRNA on the top of Suv39h dKO DE to test if H3K9me3 and DNA methylation is affected or not, and also examine whether IAP is depressed or not.

We have included the Suv39h dko analysis in new Supplementary Figure 8. We conclude that although Suv39h dko cells lose H3K9me3 from satellite repeats, there are minor effects on ERVs. In particular, for IAP elements there is no loss of H3K9me3 detectable and expression of IAP elements is not altered in Suv39h dko DE cells.

To assess whether Suv39h enzymes could compensate for the loss of Setdb1 in DE cells, we tried two approaches: (1) inhibitor treatment of Suv39h enzymes using chaetocin; (2) siRNA for Suv39h enzymes in Setdb1^{END} DE cells.

Chaetocin is an established inhibitor for Suv39h enzymes, and we could detect overall reduced H3K9me3 in wild type ES cells upon chaetocin treatment. Unfortunately, upon chaetocin treatment during DE differentiation, we did not observe reduced H3K9me3 on either IAP elements or Satellite Repeats by CHIP-seq. As H3K9me3 was not lost from satellite repeats either (unlike in the Suv39h dko DE cells), we concluded that chaetocin is not very effective in DE differentiation.

For the second strategy, we tested several Suv39h siRNAs in ES cells, where we could reduce Suv39h expression to <20%. Unfortunately, after a lot of establishment work, it turned out that siRNA transfection is quite toxic in the context of DE differentiation. We could only isolate very few DE cells that we subjected to RT-qPCR for IAP expression. We did not detect IAP up-regulation in Setdb1^{END}/Suv39h siRNA treated DE cells. These preliminary data could suggest that Suv39h enzymes do not compensate for the loss of Setdb1 in DE cells and that other HMTases would do this job. However, we can also not exclude that the knock-down efficiency was too low to detect measurable effects. We also re-analyzed the H3K9me3 CHIP-seq data from Nicetto et al., 2019 who investigated

H3K9me3 in control vs Setdb1; Suv39h triple ko liver cells. This analysis revealed that H3K9me3 is not fully lost on IAPeZ elements in triple ko cells, strongly suggesting additional H3K9me3-specific enzymes in maintaining H3K9me3 in definitive endoderm derived cell types (Addendum 1)

Therefore, we think a much more systematic approach using genetic deletion would be needed to assess this compensation mechanisms and to identify which histone methyltransferase(s) could compensate the loss of Setdb1 in DE cells. This approach would require combining Setdb1 conditional knock-out with knock-out alleles of different H3K9-specific histone methyltransferases (Suv39h, Setdb2, Prdm enzymes). We think this would be an entirely different project. Although the ontogenetic differences in establishing H3K9me3 are very interesting, we wanted to focus our study more towards the point that H3K9me3, the hallmark of heterochromatin, is not sufficient for silencing and might rather help in maintaining DNA methylation. Due to the specific effects of Setdb1 and absence of compensation for H3K9me3, the XEN differentiation system demonstrates this point very clearly.

3. Sup Fig 3A, B

Related to the issue of 2,

The authors argued that Setdb1 KO in DE cells does not diminish the H3K9me3 levels on IAP. However, % of GFP positive population between the in vitro differentiation culture for XEN- and DE-lineaged cells are different (88.4% for XEN and 44.4% for DE) and Setdb1 mRNA is still detected in DE cells but not XEN cells, suggesting that timing of Setdb1 deletion might be different between two populations. If Setdb1 depletion occurs later timing in DE cells than XEN cells, this comparison is not appropriate. Even though SETDB1 is similarly depleted in XEN and DE cells at the analyzed time point (Fig. S3C), the authors should check timing of SETDB1 depletion in this in vitro differentiation culture and compare at more appropriate timing if timing of SETDB1 depletion is different between two culture conditions.

The reviewer points out that timing of Setdb1 deletion kinetics might be different between DE and XEN cells. To address this point, we performed a time-series for Setdb1 deletion in DE vs XEN differentiation (new Supplementary Figure 3H). We find that Setdb1 is depleted to similar extent at day 7 of analysis, however, robust deletion was also detected in XEN cells at day 5 of differentiation. This indicates that deletion is 1-2 days faster in XEN cells. However, when we analyzed DE cells at day 9, which would be more equivalent to XEN cells at day 7 in terms of deletion time, we could still not detect IAP de-repression. These data confirm our previous results that Setdb1 is critical for IAP silencing only in XEN, not in DE cells.

Minor comments

1. The authors generalized derepression of IAP is restricted to VE and does not occur in DE using mouse model and in vitro differentiated XEN and DE cells derived from Setdb1^{END} ESCs. The authors clearly showed that the difference of IAP derepression between Setdb1 KO XEN and DE cells. However, in vivo part it is not clear. For example, why is the red signal increased even in extraembryonic part in the GAG staining in Fig. 2C ? Also, in Fig. 2D, the authors indicated AFP positive cells by white arrowheads. However, the GAG staining signal is increased in the entire region of Setdb1^{END} embryo.

The reviewer is correct in pointing out that the extraembryonic endoderm region of Setdb1^{END} embryos shows clear IAP de-repression. This is consistent with our line of argumentation that VE-derived cells show de-repression, but DE cells do not. Due to the presence of some cells with IAP GAG signal in the region of the embryo, where definitive endoderm cells predominate (shown by the

dotted line) we wondered if these cells are of VE origin. In absence of a lineage tracer mouse model, we simply asked whether cells in the DE region that have IAP GAG staining are positive for Afp, indicating their VE origin. During development, the VE-derived cells in the DE region of the embryo lose Afp expression and assume a DE transcription program. Therefore, it is only possible to detect these cells at an early stage during embryonic development. Our data show that VE-derived cells in the DE region of the embryo display IAP GAG staining and demonstrate that IAP-GAG positive cells within the definitive endoderm regions of embryos (which is otherwise largely negative for IAP-GAG) are likely to be derived from AFP-positive extraembryonic endoderm cells.

2. In Fig. 4, Is it possible to do SETDB1 ChIP experiments to examine whether SETDB1 is localized on IAP locus both in XEN cells and DE cells? This exp also clarify whether SETDB1 is not involved in H3K9me3 on IAP in DE cells.

To clarify whether Setdb1 is involved in H3K9me3 on IAP in DE cells, we performed ChIP-seq using a Setdb1 3xFLAG knock-in allele. We included these data as new Supplementary Figure 7 and demonstrate that Setdb1 binds IAP elements in DE cells. These data support the idea that Setdb1 is involved in mediating H3K9me3 in DE, but that loss of Setdb1 is compensated by other histone methyltransferases.

3. Fig. 5

DNA methylation status on ERVs should be shown in control and Dnmt1 KO ES, XEN and DE cells

We have performed DNA methylation analyses in Dnmt1 ko cells. These new data are shown in new Supplementary Figure 10. They show that absence of Dnmt1 results in a dramatic reduction in DNA methylation in ES cells, DE and XEN cells.

3. Supplementary Fig. 5

Figures A-C are missing

We probably made a mistake in the initial submission and forgot to upload these figures. We have now included them in the revised manuscript.

Reviewer #2 (Remarks to the Author):

The authors demonstrated that conditional knock-out of the H3K9me3 HMTase Setdb1 in mouse embryonic endoderm results in ERV de-repression in visceral endoderm (VE) descendants but not in definitive endoderm (DE), suggesting differential regulatory mechanisms in endoderm cells depending on their developmental origin. Deletion of Setdb1 resulted in loss of H3K9me3 and reduced DNA methylation of IAP elements specifically in VE progenitors. In contrast, Dnmt1 knock-out resulted in ERV de-repression in both visceral and definitive endoderm cells without affecting H3K9me3 marks. Based on these findings, the authors concluded that Setdb1-mediated H3K9me3 is not sufficient for ERV silencing, but rather critical for maintaining high DNA methylation.

Although the experimental data are solid, differential dependency of ERV silencing on Setdb1-mediated H3K9me3 in somatic cells has already been reported by other groups. Several developing organs show ERV derepression in Setdb1 KO mice, but somatic cells in adults are rather resistant to Setdb1 deletion (Tan et al., *Development* 2012, Koide et al., *Blood* 2016; Kato et al., *Nat Commun* 2018). The finding that Dnmt1 knock-out induces ERV de-repression in endoderm cells without

affecting H3K9me3 marks is interesting. This finding contrasts with the ERV silencing in ESCs, and highlights the essential role for DNA methylation in somatic cells. As the authors proposed, Setdb1-mediated H3K9me3 may function in maintaining high DNA methylation in somatic cells. In this regard, the underlying molecular mechanism is quite important. How does Setdb1-mediated H3K9me3 protect the DNA methylation levels? Additional experimental data which provides the functional link between Setdb1-mediated H3K9me3 and DNA methylation would improve the impact of this study.

We thank the reviewer for judging our finding that ERVs can be de-repressed without altering H3K9me3 as interesting. Obviously, the mechanisms for how exactly H3K9me3 helps to maintain DNA methylation, and how DNA methylation ensures silencing emerge as important questions in the context of transcriptional silencing. These mechanisms are likely to be very complex and will require follow-up studies. The reason for the assumed complexity is as follows. DNA methylation is maintained by Dnmt1, which works together with Uhrf1. Uhrf1 is assumed to target Dnmt1 and features a H3K9me3 binding domain. So, the easiest assumption would be to hypothesize that Uhrf1 binds H3K9me3 regions to target Dnmt1 for methylation maintenance. However, this hypothesis was already challenged, as it turned out that DNA methylation is maintained in a Uhrf1 mutant lacking the H3K9me3 binding domain (Zhao et al., 2016, PMC5426519). So additional studies would need to investigate a possible compensation by Uhrf2. In addition, DNA methylation dynamics is regulated by antagonizing Tet enzymes which induce DNA demethylation. Based on the complexity of the system with antagonizing activities of DNA methyltransferases and helper proteins vs. DNA demethylation machinery, follow-up studies are clearly needed to better understand the mechanistic basis for the connection between H3K9me3 and maintained DNA methylation in the context of transcriptional repression.

Other concerns are as follows.

1. Two H3K9me3 HMTases, Setdb1 and Suv39h, are assumed to function differentially as well as redundantly in ERV silencing. To understand their roles in ERV silencing, ChIP-seq of Setdb1 and Suv39h or profiling of ERV expression in Suv39h KO in comparison with Setdb1 KO would be informative.

To assess IAP binding of Setdb1 we generated ChIP-seq profiles using our 3xFLAG tagged Setdb1 knock-in cell line. The profiles in ES cells and DE cells are shown in new Supplementary Figure 7 and demonstrate that Setdb1 is enriched with these elements in both cell types. Unfortunately, we were not able to perform Suv39h ChIP-seq profiling due to technical difficulties. The only published profiles for Suv39h enzymes are from Bulut-Karslioglu et al., 2014; PMID: 24981170 from FLAG/HA tagged ES cell lines and, to our knowledge, no antibodies exist to ChIP endogenous Suv39h enzymes. In their paper, Bulut-Karslioglu et al show that in ES cells Suv39h1/2 enzymes associate with IAP elements. Therefore, both Setdb1 and Suv39h enzymes can associate with IAP elements and may also do so in DE cells. To clarify if Suv39h enzymes affect IAP silencing in DE cells we profiled expression, H3K9me3 and DNA methylation in control vs. Suv39h dko DE cells. The data are shown in new Supplementary Figure 8. We did not detect strong changes in ERV expression in either ES cells or DE cells, suggesting that Suv39h enzymes have minor roles in ERV regulation. We also failed to detect strong changes in H3K9me3 on ERVs but detected clearly reduced H3K9me3 on major satellite repeats (GSAT_MM) which are the major targets for Suv39h enzymes. Based on these analyses we would conclude that Setdb1 has predominant roles for ERV silencing in ES cells (Karimi et al. 2011), DE and XEN cells (our data), whereas Suv39h enzymes do not seem to affect ERV regulation strongly in these cell types. Regarding IAP elements, we did not detect loss of H3K9me3 in Setdb1^{END} DE cells, suggesting compensation by other H3K9me3 HMTases. As several HMTases can mediate H3K9me3

(Suv39h1/2, Setdb2, different PRDM enzymes), identification of the compensation mechanism in DE cell will require follow-up analyses.

2. Please show DNA methylation levels in Dnmt1 KO XEN and DE cells. Are they really low in Dnmt1 KO in endodermal cells?

We have now included DNA methylation analysis in Dnmt1 knock-out cells (new Supplementary Figure 10). The data clearly show that DNA methylation is strongly reduced in these cells.

3. Why is Nepn, an important regulator of endoderm development downregulated in Setdb1^{END} embryos?

Nepn downregulation is likely a secondary effect, as also suspected by reviewer 3.

Nepn is widely used as a specific marker to identify the mouse midgut region and its expression is regulated by several genes and signaling pathways involved in endoderm development (PMCID: PMC4197584). The exact function of Nepn in endoderm development is still not clear. Nepn functions as an inhibitor of TGF- β signaling, a key regulator of endoderm development, suggesting a regulatory role in endoderm development (PMID: 16990280). One recent paper showed ESCs lacking Nepn affect endoderm differentiation with reduction of endoderm markers, Gata4 and Gata6, but Nepn mutant mice could develop normally and are fertile (PMCID: PMC6981620). In the light of these studies, I think Nepn is more likely to serve as a sensor for malformation of endoderm development, and the strong reduction in Nepn expression indicative of defective endoderm development.

Reviewer #3 (Remarks to the Author):

H3K9me3 is a mark of constitutive heterochromatin, and is tightly correlated with high DNA methylation. In mammals, several H3K9 methyltransferases exist, and it is not fully resolved what their specific and redundant activities are. Moreover, it is not clear what the epistatic relationship is between the histone mark and DNA methylation-based regulation. In mice, particularly mouse embryonic stem cells, the H3K9 methyltransferase SETDB1 is associated with silent endogenous retroviruses (ERVs), markedly IAPs. Given cell-type specific differences of ERV regulation, here Wang et al set out to explore the role of SETDB1 in both in vivo and in vitro models of endoderm differentiation. Using an endoderm-specific inducible knockout system, they made the interesting finding that SETDB1 appears to be important for regulating ERVs in extra embryonic endoderm (akin to its role in ESCs), and less so in definitive endoderm. The authors expounded in this finding by employing an in vitro endoderm differentiation system, consistently showing a requirement for SETDB1 in the extra embryonic (XEN) rather than embryonic (DE) cell types. Finally, the authors attempt to untangle the relationship between H3K9 and DNA methylation using a Dnmt1 mutant. Whereas DNA methylation plays an important role in repression of ERVs in both XEN and DE cells, surprisingly H3K9me3 is unaffected at the IAPez family of ERVs. This indicates that DNA methylation is the more downstream effector of silencing of ERVs, and is an intriguing result. I should say that I remain partially skeptical about this result (see below).

Overall, the paper is concise and clearly expounded. Moreover, I found the combination of elegant mouse genetics with an in vitro system to be a strong validation of their results. However, I did find

the paper to be somewhat deficient in some areas. While the finding that SETDB1 has cell type-specific effects is interesting and worth publishing, per se, there is very little experimental digging into the how (see major point) or the why. What is the biological underpinning for these changes during development, and why would there be such a stark difference between embryonic and extra embryonic tissues? As it stands, the paper is missing these key elements to push it into a journal such as Nature Communications.

Major Points

• The authors make an important finding that SETDB1 only has an importance in extra-embryonic endoderm tissues, however they never explore the genetic interactions between the various HMTs. I understand that performing further genetic tests in the complicated in vivo system would be very time consuming, however they have implemented an ESC system where these experiments should be relatively straightforward to achieve. I suggest comparing *Setdb1*, *suvr39* dKO, and the triple knockout cells in parallel. A conditional knockout (or degron) system could be utilized in case the triple knockout is not viable. As the authors mentioned, HMTase tKO data was published recently (Nicetto et al, 2019). However, I believe the authors could truly expand upon their findings by exploring the effects on histone methylation, DNA methylation, and expression in combinatorial mutant lines. Partially related, I would have liked to have seen more analyses presented from the ChIP-Seq data. Were there any regions where H3K9me3 was impacted in the *Setdb1* mutant line? The authors only present data from retrotransposons, and do not discuss other genomic compartments in the text. This could be more flushed out with increased results in other genetic backgrounds.

To assess the compensation mechanism in DE cells, the reviewer suggests to investigate H3K9me3 and DNA methylation in *Setdb1*, *Suv39h* dko and triple ko DE cells. To address this, we have now included the *Suv39h* dko analysis in new Supplementary Figure 8. We conclude that although *Suv39h* dko cells lose H3K9me3 from satellite repeats, there are minor effects on ERVs. For IAP elements there is no loss of H3K9me3 detectable and expression of IAP elements is not altered in *Suv39h* dko DE cells.

Establishing triple ko ES cells or degron cells, as suggested by the reviewer requires a lot of time. Due to the Covid pandemic we actually had a long delay in getting the first author back from China to the lab, and he could only spend a few month doing hands on work for the revision. Therefore, we tried to assess compensation mechanism by *Suv39h* enzymes with less time-consuming approaches: (1) inhibitor treatment of *Suv39h* enzymes using chaetocin; (2) siRNA for *Suv39h* enzymes in *Setdb1*^{END} DE cells.

Chaetocin is an established inhibitor for *Suv39h* enzymes, and we could detect overall reduced H3K9me3 in wild type ES cells upon chaetocin treatment. Unfortunately, upon chaetocin treatment during DE differentiation, we did not observe reduced H3K9me3 on either IAP elements or Satellite Repeats by ChIP-seq. As H3K9me3 was not lost from satellite repeats (unlike in the *Suv39h* dko DE cells), we concluded that chaetocin is not very effective in DE differentiation.

For the second strategy, we tested several *Suv39h* siRNAs in ES cells, where we could reduce *Suv39h* expression to <20%. Unfortunately, after a lot of establishment work, it turned out that siRNA transfection is quite toxic in the context of DE differentiation. We could only isolate very few DE cells that we subjected to RT-qPCR for IAP expression. We did not detect IAP up-regulation in *Setdb1*^{END}/*Suv39h* siRNA treated DE cells. These preliminary data suggest that *Suv39h* enzymes do not compensate for the loss of *Setdb1* in DE cells and that other HMTases would do this job. However, we can also not exclude that the knock-down efficiency was too low to detect measurable

effects. Therefore, we think a much more systematic approach using genetic deletion would be needed to assess this compensation mechanisms and to identify which histone methyltransferase(s) could compensate the loss of Setdb1 in DE cells. This approach would actually require combining Setdb1 conditional knock-out with knock-out alleles of different H3K9-specific histone methyltransferases (Suv39h, Setdb2, Prdm enzymes). We think this would be an entirely different project. Although the ontogenetic differences in establishing H3K9me3 are very interesting, we wanted to focus our study more towards the point that H3K9me3, the hallmark of heterochromatin, is not sufficient for silencing and might rather help in maintaining DNA methylation. Due to the specific effects of Setdb1 and absence of compensation for H3K9me3, the XEN differentiation system demonstrates this point very clearly.

As suggested by the reviewer we also investigated the H3K9me3 data in more detail to assess regions outside ERV elements. The new analysis is shown in Supplementary Figures 4 and 5. Globally, H3K9me3 distribution is very different between DE and XEN cells, with large megabase-size domains appearing in DE cells, but largely absent in XEN cells. In Setdb1^{END} DE cells, we detect largely unaltered H3K9me3 domains. Interestingly, in Setdb1^{END} XEN cells, we detect appearance of many large H3K9me3 domains in new genomic regions. These data indicate that H3K9me3 regulation and Setdb1-dependance is very different between DE and XEN cells. We than also investigated local H3K9me3 changes in the vicinity of genes to detect H3K9me3-regulated genes. Here, we identified H3K9me3 peak regions which lose H3K9me3 in Setdb1^{END} DE and XEN cells. This analysis revealed several genes (much more in XEN cells as compared to DE) which were marked by H3K9me3 and became derepressed in Setdb1^{END} cells. Again, these data highlight that Setdb1 is important for gene silencing in both DE and XEN cells, but that H3K9me3-marked and regulated genes differ between these lineages.

- The authors make the claim that in the Dnmt1 knockout cells, expression of ERVs is higher, but H3K9me3 at IAPez remains unchanged, seen by both ChIP-qPCR and -seq. However, given that these are repetitive elements, how can one conclude that the increased expression is not due to a small number of highly expressed loci that are NOT marked for H3K9me3? These would be simply missed from the analysis. I understand this is a hard point to address without using long read sequencing technology, but it should at least be discussed as a possibility before affirming the genetic relationship. I am not aware of any biological situations where H3K9me3-marked loci are associated with such high expression, even in the case of heterochromatic RNAs. Of course, it being biology, you never know.

This is indeed a difficult point to address. We have not performed long read sequencing and therefore we could only identify H3K9me3 positive IAP elements based on H3K9me3 spreading into neighboring regions that are not repetitive. Not all IAP elements show strong H3K9me3 in their border regions, which could be associated with polymorphisms within these elements that preclude recruitment of Setdb1. To assess the transcription of individual elements we searched for H3K9me3 positive IAP insertions which do not properly terminate transcription on their downstream LTR. Indeed, we could identify examples for H3K9me3 positive IAPs with transcriptional activity. We have added an IGV screenshot to demonstrate this (new Figure 5C). This example highlights that H3K9me3 positive IAP elements can be transcriptionally active, although we cannot exclude that also H3K9me3 negative IAPs substantially contribute to the overall up-regulation of these repeats. We have discussed this now in the text.

Minor Points

- With regards to embryology, I am not an expert and it would have been useful to have a simple schematic for endoderm differentiation in the supplement, so those (such as me) would not need to refer to google.

This is a good suggestion. We have now provided a schematic to better show the important aspects of early embryonic endoderm development in new Supplementary Figure 1B.

- Abstract: A tad confusing to describe modifications on ERVs as “established” by the “maintenance” DNA methyltransferase. Perhaps use another verb, such as “deposited”

As suggested by the reviewer we have changed the verb to “deposited”.

- Line 132: Nepn downregulation is likely a secondary effect, I presume. It’s striking how much stronger its downregulation is compared to any other gene. Could the authors speculate? Does it serve as a sensor for malforming endoderm tissue?

As suspected by the reviewer, Nepn downregulation is likely a secondary effect. Nepn is widely used as a specific marker to identify the mouse midgut region and its expression is regulated by several genes and signaling pathways involved in endoderm development (PMCID: PMC4197584). The exact function of Nepn in endoderm development is still not clear. Nepn functions as an inhibitor of TGF- β signaling, a key regulator of endoderm development, suggesting a regulatory role in endoderm development (PMID: 16990280). One recent paper showed ESCs lacking Nepn affect endoderm differentiation with reduction of endoderm markers, Gata4 and Gata6, but Nepn mutant mice could develop normally and are fertile (PMCID: PMC6981620). In the light of these studies, I think Nepn is more likely to serve as a sensor for malformation of endoderm development, and the strong reduction in Nepn expression indicative of defective endoderm development.

- Line 138: Not all LINE families are unchanged. It appears one is significantly unregulated (L1Md_T|LINE|L1). This should be at least acknowledged

We have now stated this upregulation in the text.

- It is never clear how many embryos are being assessed for each experiment. For example, in Figure 2D, is the statement about VE and GAG expression being concluded from only one embryo? This information should at least be added to the figure legends.

Generally, we have used representative images from three embryos per genotype and stage. We have added this information in the respective figure legends.

- Line 211 and Figure S4B: Given that LINE K9me3 methylation is not impacted in Setdb1 mutant XEN cells, I’m surprised that DNA methylation and expression are. The authors never discuss this curiosity. What is the explanation? Perhaps it is because Setdb1 is required for recruiting de novo methylation through DNMT3A and/or B during differentiation, which clearly must occur on these elements?

Yes, this is an interesting observation to which we did not pay attention before. It is possible that Setdb1 is critical for the recruitment of Dnmts in this context, and other HMTases could establish H3K9me3. It is certainly worth to go deeper into this in follow-up studies. We have now discussed this finding in the text.

- In the PDF, Figure panels S5A-C appear to be missing, which are important control experiments.

We probably made a mistake in the initial submission and forgot to upload these figures. We have now included them in the revised manuscript.

- I would place Figure S5E in the primary, and put the CHIP qPCR in the supplemental. I found the CHIP-Seq data the more compelling result.

As suggested by the reviewer we have rearranged Figure 5.

Addendum 1

A

B

C

Addendum 1. H3K9me3 is reduced but not completely lost from IAPEz element in triple ko liver cells.

(A) Dot plot showing basemean expression vs. log₂-fold change of ERV families in control vs. Setdb1; Suv39h1; Suv39h2 triple ko liver cells. ERV families with significantly changed expression (adjusted p-value < 0.01; n=3 for each condition) are colored. Selected ERV families are labeled.

(B) Dot plot showing basemean expression vs. log₂-fold change of LINE families in control vs. Setdb1; Suv39h1; Suv39h2 triple ko liver cells. LINE families with significantly changed expression (adjusted p-value < 0.01; n=3 for each condition) are colored. Selected LINE families are labeled.

(C) Dot plot showing expression vs. H3K9me3 changes on ERV families between control and Setdb1; Suv39h1; Suv39h2 triple ko liver cells. ERV families with significantly changed expression (fold change > 2; n=3 for each condition) are colored. Selected ERV families are labeled.

Cumulative H3K9me3 CHIP-seq coverage across IAP elements in control and Setdb1; Suv39h1; Suv39h2 triple ko liver cells. The structure of IAP elements is shown schematically. (rpkm = reads per kilobase per million of reads).

REVIEWER COMMENTS

Reviewer #1 (Remarks to the Author):

The Minor comments, the authors revised appropriately. However, the issues of major comments are still not solved much, especially dispensability of SETDB1 for H3K9me3 and silencing of IAPs.

The authors responded to the issue 3 of major comment (timing of SETDB1 depletion might be different between XEN and DE), but new data is still not sufficient to conclude that “In DE, loss of Setdb1 did not affect H3K9me3 nor DNA methylation, suggesting Setdb1-independent pathways for maintaining these modifications.” P2, line 40-41. The authors examined the induced DE cells at a longer time period post Setdb1 deletion (d7 in Fig. S3H) and found same results. However, IAP derepression in XEN was also only induced d7 (but not d5) post Setdb1 depletion and this is two days after almost complete depletion of Setdb1 mRNA signals (d5). From the Fig. S3H data, the reviewer expected that d9 (not d7) DE is equivalent to d5 XEN for Setdb1 mRNA depletion. Therefore, the authors should use d11 (2 days after complete Setdb1 mRNA depletion) DE cells for IAP depression and H3K9me3 studies. If cell division time is different (DE grows slower than XEN), passive (replication dependent) H3K9m3 reduction may take more time for DE and >d11 cells may be needed. Without such data, it is difficult for the reviewer to accept their major conclusion for the dispensability of SETDB1 in H3K9me3 and repression of IAPs in DE cells and the potential function of SETDB1 for “maintaining high DNA methylation”.

Reviewer #2 (Remarks to the Author):

The authors did not address my major concern; the molecular mechanism underlying maintenance of DNA methylation by Setdb1-mediated H3K9me3 in visceral endoderm. Since differential dependency of ERV silencing on Setdb1-mediated H3K9me3 in somatic cells has already been reported by other groups, the lack of mechanistic insights impair the impact of this manuscript. Otherwise, the authors precisely addressed my concerns.

Reviewer #3 (Remarks to the Author):

The authors thoughtfully and diligently responded to my comments. I also sympathize with pandemic-related difficulty in performing certain experiments. I believe the rebuttal was satisfactory, and I have no reservations recommending for publication.

REVIEWER COMMENTS

Reviewer #1 (Remarks to the Author):

The Minor comments, the authors revised appropriately. However, the issues of major comments are still not solved much, especially dispensability of SETDB1 for H3K9me3 and silencing of IAPs. The authors responded to the issue 3 of major comment (timing of SETDB1 depletion might be different between XEN and DE), but new data is still not sufficient to conclude that "In DE, loss of Setdb1 did not affect H3K9me3 nor DNA methylation, suggesting Setdb1-independent pathways for maintaining these modifications." P2, line 40-41. The authors examined the induced DE cells at a longer time period post Setdb1 deletion (d7 in Fig. S3H) and found same results. However, IAP derepression in XEN was also only induced d7 (but not d5) post Setdb1 depletion and this is two days after almost complete depletion of Setdb1 mRNA signals (d5). From the Fig. S3H data, the reviewer expected that d9 (not d7) DE is equivalent to d5 XEN for Setdb1 mRNA depletion. Therefore, the authors should use d11 (2 days after complete Setdb1 mRNA depletion) DE cells for IAP depression and H3K9me3 studies. If cell division time is different (DE grows slower than XEN), passive (replication dependent) H3K9me3 reduction may take more time for DE and >d11 cells may be needed. Without such data, it is difficult for the reviewer to accept their major conclusion for the dispensability of SETDB1 in H3K9me3 and repression of IAPs in DE cells and the potential function of SETDB1 for "maintaining high DNA methylation".

We are thankful to the reviewer for acknowledging that we largely revised the manuscript appropriately.

The reviewer is still concerned about our finding that Setdb1^{END} cells do not lose H3K9me3 on IAPEz elements in DE cells. One possible explanation is that passive loss of H3K9me3 takes more time in DE cells as compared to XEN cells. In order to address this question, we established an extended DE differentiation time-course until day 14 of differentiation. This time-course covers 5 more days as before and around 7 days after Setdb1 is largely deleted. Due to the intricacies of the Cre/lox system we cannot get a 100% knock-out as we only monitor Cre activity with the EGFP reporter and a certain percentage of cells can escape Setdb1 deletion. We monitored IAPEz expression at day 8, 10, 12 and 14 and could not observe noticeable de-repression (new Supplementary Figure 7B). We agree with the reviewer that proliferation can be a critical factor for passive loss of histone modifications upon HMTase removal. Therefore, we controlled for proliferation of DE cells over the extended time-course by Hoechst staining for DNA content followed by FACS analysis (new Supplementary Figure 7C). We could clearly detect that DE cells proliferate over the entire time course. Only at day 14 we noticed that both control and Setdb1^{END} DE cells had reduced fitness and slightly reduced proliferation. We think that DE cells, for unknown reasons, cannot be kept much longer under these culturing conditions. Therefore, we did not extend the time-course beyond day 14. The cell cycle analysis clearly shows that Setdb1^{END} cells proliferate in absence on Setdb1, which should allow passive loss of H3K9me3 on Setdb1 sensitive regions. To assess changes in H3K9me3 on IAPEz elements we performed new ChIP-seq experiments on cells at day 12 (n=3) and day 14 (n=2) of the extended DE differentiation time-course. We could not detect reduced H3K9me3 on IAPEz elements on

either day 12 or day 14 (new Supplementary Figure 7D). We would argue that our new data are compatible with the interpretation that Setdb1 loss is compensated by other HMTases in DE differentiation and hope that the reviewer agrees with our line of argumentation.

Reviewer #2 (Remarks to the Author):

The authors did not address my major concern; the molecular mechanism underlying maintenance of DNA methylation by Setdb1-mediated H3K9me3 in visceral endoderm. Since differential dependency of ERV silencing on Setdb1-mediated H3K9me3 in somatic cells has already been reported by other groups, the lack of mechanistic insights impair the impact of this manuscript. Otherwise, the authors precisely addressed my concerns.

We thank the reviewer for acknowledging that we could properly address most concerns. We fully agree with the reviewer that it would be extremely interesting to address and explain the molecular mechanism underlying maintenance of DNA methylation by Setdb1-mediated H3K9me3 in visceral endoderm. However, for the time and resources required, we see this enterprise very difficult in the context of this manuscript but rather as a follow-up project.

Reviewer #3 (Remarks to the Author):

The authors thoughtfully and diligently responded to my comments. I also sympathize with pandemic-related difficulty in performing certain experiments. I believe the rebuttal was satisfactory, and I have no reservations recommending for publication.

We thank the reviewer for acknowledging that we satisfactorily addressed all concerns.

REVIEWER COMMENTS

Reviewer #1 (Remarks to the Author):

In response to the reviewer's concern, the authors performed H3K9me3 ChIP-seq analysis using WT and Setdb1^{END} cells 12 and 14 days after the induction of DE cell differentiation from ES. The result was the same as the result on DE cells 7 days after differentiation induction, and the level of H3K9me3 of IAP was not decreased by Setdb1^{END}. They also provided FACS analysis stained with Hoechst as Suppl Fig. 7C. However, this Hoechst staining data simply showed that cells with different cell cycles were present at the moment of analysis, but did not demonstrate that the cells continued to proliferate up to day 12 or 14 when the cells were harvested for analysis. In addition, there is no point in comparing WT and Setdb1^{END} cells, and the authors have to compare Setdb1^{END} XEN and DE cells. Therefore, although the reviewer has no doubt about the H3K9me3 ChIP-seq result itself, it has not yet been determined whether the DE cells analyzed was equivalent to the XEN cells (enough cell divisions for depletion of SETDB1-dependent H3K9me3 progressed after Setdb1 KO). 1) Why not simply count the number of cells continuously to find out how much the cells replicate? 2) Even if cell proliferation is continuously observed, it should be shown depletion of SETDB1 from the day 12 and day14 DE cells by Western blot analysis. The issue of depletion timing of SETDB1 in DE cells is not solved, it is difficult for the reviewer to accept the authors proposal "These data suggest that Setdb1 is the major H3K9me3 HMTase for IAPeZ elements in XEN cells and, that other HMTases compensate for the loss of Setdb1 in DE cells to maintain H3K9me3 on IAPeZ elements and other ERVs." which is most unpredicted and potentially interesting finding.

If the authors can provide convincing data about the replication issue, the reviewer has no more critical comment.

REVIEWER COMMENTS

Reviewer #1 (Remarks to the Author):

In response to the reviewer's concern, the authors performed H3K9me3 ChIP-seq analysis using WT and Setdb1^{END} cells 12 and 14 days after the induction of DE cell differentiation from ES. The result was the same as the result on DE cells 7 days after differentiation induction, and the level of H3K9me3 of IAP was not decreased by Setdb1^{END}. They also provided FACS analysis stained with Hoechst as Suppl Fig. 7C. However, this Hoechst staining data simply showed that cells with different cell cycles were present at the moment of analysis, but did not demonstrate that the cells continued to proliferate up to day 12 or 14 when the cells were harvested for analysis. In addition, there is no point in comparing WT and Setdb1^{END} cells, and the authors have to compare Setdb1^{END} XEN and DE cells. Therefore, although the reviewer has no doubt about the H3K9me3 ChIP-seq result itself, it has not yet been determined whether the DE cells analyzed was equivalent to the XEN cells (enough cell

divisions for depletion of SETDB1-dependent H3K9me3 progressed after Setdb1 KO). 1) Why not simply count the number of cells continuously to find out how much the cells replicate? 2) Even if cell proliferation is continuously observed, it should be shown depletion of SETDB1 from the day 12 and day14 DE cells by Western blot analysis. The issue of depletion timing of SETDB1 in DE cells is not solved, it is difficult for the reviewer to accept the authors proposal "These data suggest that Setdb1 is the major H3K9me3 HMTase for IAPEz elements in XEN cells and, that other HMTases compensate for the loss of Setdb1 in DE cells to maintain H3K9me3 on IAPEz elements and other ERVs." which is most unpredicted and potentially interesting finding.

If the authors can provide convincing data about the replication issue, the reviewer has no more critical comment.

We thank the reviewer for the thoughtful comments. In fact, we could not fully assess how many rounds of replication DE cells have undergone after Setdb1 depletion. The reason is that the DE differentiation system is very complicated, and cells cannot easily be manipulated and followed. For the initial time course of 9 days, we kept cells unperturbed on the plate until analysis. This allowed initiation of differentiation (as seen by GFP⁺ cells) and expansion of these cells (as seen by colonies forming, an example is shown in Figure 3B). For the longer time-course we needed to split the cells to prevent the plate from becoming confluent, which would prevent proliferation. This turned out to be very tricky as the cells do not like to be taken off the plate. Ultimately, we found that mild accutase mediated dissociation and seeding on coated plates allowed the cells to be kept in culture for additional days (until day 14). For the splitting we did not separate GFP⁺ from GFP⁻ cells, therefore we could not simply count the number of GFP⁺ cells during this time-course. To address the reviewer's comment, we now tried to sort GFP⁺ cells to better monitor their subsequent proliferation. After extensive testing of conditions, we have to conclude that sorting GFP⁺

cells is not compatible with their subsequent culturing. The cells are stressed too much and die within few days after sorting (both, control cells and ko cells, so this is not a phenotype but rather a property of DE cells).

Altogether we performed several rounds of DE differentiation coupled with H3K9me3 and IAPEz transcription analysis. In none of the experiments we could find a hint for a major effect on H3K9me3 maintenance or silencing of IAPEz elements. The data are also consistent with the *in vivo* effect of *Setdb1* deletion, where we observed IAPEz derepression only in visceral endoderm (corresponding to XEN cells), but not in definitive endoderm (corresponding to DE cells). However, since the limitations of the DE system do not allow us to fully exclude that *Setdb1*-deficient DE cells undergo enough rounds of replication to passively lose all *Setdb1*-dependent H3K9me3 before analysis, we decided to highlight this limitation in the manuscript, so that readers are clearly aware of this. We therefore included a paragraph “Limitations of this study” in the main text, where we explain this limitation. We think this is a good-practice strategy that is now increasingly being used in high-quality journals (for example Barral et al., 2022, Mol Cell, PMID 35081363).

Although the finding that *Setdb1* seems to have minor roles for H3K9me3 in DE cells is surprising, one of the major findings of our manuscript is that H3K9me3 is fully maintained on transcriptionally active IAP elements in absence of DNA methylation. The point here is not that DNA methylation has a role for ERV silencing in differentiated cells. This is known for a long time. The point is that nobody (to our knowledge) has investigated the interplay between DNA methylation and H3K9me3 pathways for ERV silencing. In differentiated cells, people have looked at either *Setdb1*-mediated H3K9me3, or at DNA methylation. We find in XEN cells that loss of *Setdb1* or *Dnmt1* result in IAPEz derepression, a situation which was not described in other cell types. The importance here is the following. H3K9me3 is thought to be recognized by several adapter proteins (e.g. HP1) that mediate silencing by establishing a compact, less accessible chromatin state which is not permissive to transcriptional activation. DNA methylation, instead, is thought to prevent binding of transcription factors that are sensitive to DNA methylation in their binding site. Now, in *Dnmt1* ko XEN cells, H3K9me3 is fully present on de-repressed IAPEz elements. This means that the silencing potential of H3K9me3 through recruitment of binding factors should still be intact, but obviously it is not. Therefore, our findings suggest an upstream role of H3K9me3 for maintaining full DNA methylation and provides a paradigm to better study the chromatin-based mechanism of IAPEz silencing (and maybe of other H3K9me3/DNAme enriched regions of the genome) in future studies.

REVIEWERS' COMMENTS

Reviewer #1 (Remarks to the Author):

The reviewer's 2nd comments.

First of all, the reviewer has no further comment on the issue which the reviewer requested to clarify.

Second, the authors pointed out that "Although the finding that Setdb1 seems to have minor roles for H3K9me3 in DE cells is surprising, one of the major findings of our manuscript is that H3K9me3 is fully maintained on transcriptionally active IAP elements in absence of DNA methylation. The point here is not that DNA methylation has a role for ERV silencing in differentiated cells. This is known for a long time. The point is that nobody (to our knowledge) has investigated the interplay between DNA methylation and H3K9me3 pathways for ERV silencing. In differentiated cells, people have looked at either Setdb1-mediated H3K9me3, or at DNA methylation." The reviewer agrees with the authors above point. Furthermore, no one has investigated the mechanism that SETDB1 and SETDB1-mediated H3K9me3 does not have a role for silencing of IAP (IAPEZ) in differentiated cells such as iMEFs. The reviewer also does not remember any report described the status of H3K9me3 on the de-repressed IAPEZ in Dnmt1-KO somatic cells. However, the reviewer has already stated in the first review comments that "However, it is already shown that DNA methylation is essential but SETDB1 and SETDB1-mediated H3K9me3 is dispensable for IAP silencing in other differentiated cells (PMCID:PMC5923290). Furthermore, H3K9me3 is maintained in Setdb1 KO DE cells, still possible that H3K9me3 play additive role for IAP silencing in DE cells other than DNA-methylation pathway. In ESCs, it is suggested that both H3K9me3 and DNA methylation contribute to IAP silencing (PMCID:PMC3857791)". Furthermore, in the first cited work, they examined impact of Setdb1 KO, Dnmt1 KD and dual treatment, and found that Setdb1 KO has no impact (no additive role, too) on IAP silencing in iMEFs (Fig. 4b). Therefore, if the authors want to stress their interpretation in the revised manuscript, these introduced work should be cited properly.

REVIEWER COMMENTS

Reviewer #1 (Remarks to the Author):

The reviewer's 2nd comments.

First of all, the reviewer has no further comment on the issue which the reviewer requested to clarify.

Second, the authors pointed out that "Although the finding that Setdb1 seems to have minor roles for H3K9me3 in DE cells is surprising, one of the major findings of our manuscript is that H3K9me3 is fully maintained on transcriptionally active IAP elements in absence of DNA methylation. The point here is not that DNA methylation has a role for ERV silencing in differentiated cells. This is known for a long time. The point is that nobody (to our knowledge) has investigated the interplay between DNA methylation and H3K9me3 pathways for ERV silencing. In differentiated cells, people have looked at either Setdb1-mediated H3K9me3, or at DNA methylation." The reviewer agrees with the authors above point. Furthermore, no one has investigated the mechanism that SETDB1 and SETDB1-mediated H3K9me3 does not have a role for silencing of IAP (IAPEZ) in differentiated cells such as iMEFs. The reviewer also does not remember any report described the status of H3K9me3 on the de-repressed

IAPEZ in Dnmt1-KO somatic cells. However, the reviewer has already stated in the first review comments that "However, it is already shown that DNA methylation is essential but SETDB1 and SETDB1-mediated H3K9me3 is dispensable for IAP silencing in other differentiated cells (PMCID:PMC5923290). Furthermore, H3K9me3 is maintained in Setdb1 KO DE cells, still possible that H3K9me3 play additive role for IAP silencing in DE cells other than DNA-methylation pathway. In ESCs, it is suggested that both H3K9me3 and DNA methylation contribute to IAP silencing (PMCID:PMC3857791)". Furthermore, in the first cited work, they examined impact of Setdb1 KO, Dnmt1 KD and dual treatment, and found that Setdb1 KO has no impact (no additive role, too) on IAP silencing in iMEFs (Fig. 4b). Therefore, if the authors want to stress their interpretation in the revised manuscript, these introduced work should be cited properly.

We thank the reviewer for his insightful comments. We have now added the missing citation in the introduction.